# MMD-FUSE: Learning and Combining Kernels for Two-Sample Testing Without Data Splitting

**Felix Biggs**[*]
Centre for Artificial Intelligence
Department of Computer Science
University College London & Inria London
`contact@felixbiggs.com`

**Antonin Schrab**[*]
Centre for Artificial Intelligence
Gatsby Computational Neuroscience Unit
University College London & Inria London
`a.schrab@ucl.ac.uk`

**Arthur Gretton**
Gatsby Computational Neuroscience Unit
University College London
`arthur.gretton@gmail.com`

## Abstract

We propose novel statistics which maximise the power of a two-sample test based on the Maximum Mean Discrepancy (MMD), by adapting over the set of kernels used in defining it. For finite sets, this reduces to combining (normalised) MMD values under each of these kernels via a weighted soft maximum. Exponential concentration bounds are proved for our proposed statistics under the null and alternative. We further show how these kernels can be chosen in a data-dependent but permutation-independent way, in a well-calibrated test, avoiding data splitting. This technique applies more broadly to general permutation-based MMD testing, and includes the use of deep kernels with features learnt using unsupervised models such as auto-encoders. We highlight the applicability of our MMD-FUSE test on both synthetic low-dimensional and real-world high-dimensional data, and compare its performance in terms of power against current state-of-the-art kernel tests.

## 1  Introduction

The fundamental problem of non-parametric two-sample testing consists in detecting the difference between any two distributions having access only to samples from these. Kernel-based tests relying on the Maximum Mean Discrepancy (MMD; Gretton et al., 2012a) as a measure of distance on distributions are well-suited for this framework as they can identify complex non-linear features in the data, and benefit from both strong theoretical guarantees and ease of implementation. This explains their popularity among practitioners and justifies their wide use for real-world applications.

However, the performance of these tests is crucially impacted by the choice of kernel. This is commonly tackled by either: choosing the kernel by some weakly data-dependent heuristic (Gretton et al., 2012a); or splitting off a hold-out set of data for kernel selection, with the other half used for the actual test (Gretton et al., 2012b; Sutherland et al., 2017). This latter method includes training feature extractors such as deep kernels on the first selection half. Both of these methods can incur a significant loss in test power, since heuristics may lead to poor kernel choices, and data splitting reduces the number of data points for the actual test.

Our contribution is to present MMD-based tests which can strongly adapt to the data *without* data splitting. This comes in two parallel parts: firstly we show how the kernel can be chosen in an

---

[*]Equal contribution.

37th Conference on Neural Information Processing Systems (NeurIPS 2023).

unsupervised fashion using the entire dataset, and secondly we show how multiple such kernels can be adaptively weighted in a single test statistic, optimising test power.

**Data Splitting.** The data splitting approach selects test parameters on a held-out half of the dataset, and applies the test to the other half. Commonly, this involves optimising a kernel on held-out data in a supervised fashion to distinguish which sample originated from which distribution, either by learning a deep kernel directly (Sutherland et al., 2017; Liu et al., 2020, 2021), or indirectly through the associated witness function (Kübler et al., 2022a,b). Jitkrittum et al. (2016) propose tests which select witness function features (in either spatial or frequency space) on the held-out data, running the analytic representation test of Chwialkowski et al. (2015) on the remaining data.

Our first contribution is to show that it is possible to learn a feature extractor for our test (*e.g.*, a deep kernel) on the *entirety* of the data in an *unsupervised* manner while retaining the desired non-asymptotic test level. Specifically, any method can be used that is ignorant of which distribution generated which samples. We can thus leverage many feature extraction methods, from auto-encoders (Hinton and Salakhutdinov, 2006) to recent powerful developments in self-supervised learning (He et al., 2020; Chen et al., 2020a,b,c; Chen and He, 2021; Grill et al., 2020; Caron et al., 2020; Zbontar et al., 2021; Li et al., 2021). Remarkably, our method applies to *any* permutation-based two-sample test, even non-kernel tests, provided the parameters are chosen in this unsupervised fashion. This includes a very wide array of MMD-based tests, and finally provides a formal justification for the commonly-used median heuristic (Gretton et al., 2012b; Ramdas et al., 2015; Reddi et al., 2015).

**Adaptive Kernel Selection.** A newer approach originating in Schrab et al. (2023) performs adaptive kernel selection through multiple testing. Aggregating several MMD-based tests with different kernels, each on the whole dataset, results in an overall adaptive test with optimal power guarantees in terms of minimax separation rate over Sobolev balls. Variants of this kernel adaptivity through aggregation have been proposed with linear-time estimators (Schrab et al., 2022b), spectral regularisation (Hagrass et al., 2022), kernel thinning compression (Domingo-Enrich et al., 2023), and in the asymptotic regime (Chatterjee and Bhattacharya, 2023). Another adaptive approach using the entire dataset for both kernel selection and testing is given by Kübler et al. (2020); this leverages the Post Selection Inference framework, but the resulting test suffers from low power in practice.

While our unsupervised feature extraction method is an extremely general technique and potentially powerful, particularly for high-dimensional structured data like images, it is not always sufficient for data where such feature extraction is difficult. This motivates our second contribution, a method for combining whole-dataset MMD estimates under multiple different kernels into single test statistics, for which we prove exponential concentration. Kernel parameters such as bandwidth can then be chosen in a non-heuristic manner to optimise power, even on data with less structure and varying length scales. Using a single statistic also ensures that a single *test* can be used, instead of the multiple testing approach outlined above, reducing computational expense.

**MMD-FUSE.** By combining these contributions we construct two closely related MMD-FUSE tests. Each chooses a set of kernels based on the whole dataset in an unsupervised fashion, and then adaptively weights and *fuses* this (potentially infinite) set of kernels through our new statistics; both parts of this procedure are done using the entire dataset without splitting. On the finite sets of kernels we use in practice, the weighting procedure is done in closed form via a weighted soft maximum. We show these new tests to be well-calibrated and give sufficient power conditions which achieve the minimax optimal rate in terms of MMD. In empirical comparisons, our test compares favourably to the state-of-the-art aggregated tests in terms of power and computational cost.

**Outline and Summary of Contributions.** In Section 2 we outline our setting, crucial results underlying our work, and alternative existing approaches. Section 3 covers the construction of permutation tests and discusses how we can choose the parameters for any such test in any unsupervised fashion including with deep kernels and by those methods mentioned above. Section 4 introduces and motivates our two proposed tests. Section 5 discusses sufficient conditions for test power (at a minimax optimal rate in MMD), and the exponential concentration of our statistics. Finally, we show that our test compares favourably with a wide variety of competitors in Section 6 and discuss in Section 7.

## 2 Background

**Two-Sample Testing.** The two-sample testing problem is to determine whether two distributions $p$ and $q$ are equal or not. In order to test this hypothesis, we are given access to two samples, $\boldsymbol{X} \coloneqq (X_1, \ldots, X_n) \overset{\text{iid}}{\sim} p$ and $\boldsymbol{Y} \coloneqq (Y_1, \ldots, Y_m) \overset{\text{iid}}{\sim} q$ as tuples of data points with sizes $n$ and $m$. We write the combined (ordered) sample as $\boldsymbol{Z} \coloneqq (Z_1, \ldots, Z_{n+m}) = (\boldsymbol{X}, \boldsymbol{Y}) = (X_1, \ldots, X_n, Y_1, \ldots, Y_m)$.

We define the null hypothesis $H_0$ as $p = q$ and the alternative hypothesis $H_1$ as $p \neq q$, usually with a requirement $\mathbb{D}(p, q) > \epsilon$ for a distance $\mathbb{D}$ (such as the MMD) and some $\epsilon > 0$. A hypothesis test $\Delta$ is a $\{0, 1\}$-valued function of $\boldsymbol{Z}$, which rejects the null hypothesis if $\Delta(\boldsymbol{Z}) = 1$ and fails to reject it otherwise. It will usually be formulated to control the probability of a type I error at some level $\alpha \in (0, 1)$, so that $\mathbb{P}_{p \times p}(\Delta(\boldsymbol{Z}) = 1) \leq \alpha$, while simultaneously minimising the probability of a type II error, $\mathbb{P}_{p \times q}(\Delta(\boldsymbol{Z}) = 0)$. In the above we have used the notation $\mathbb{P}_{p \times p}$ and $\mathbb{P}_{p \times q}$ to indicate that the sample $\boldsymbol{Z}$ is either drawn from the null, $p = q$, or the alternative $p \neq q$. Similar notation will be used for expectations and variances. When a bound $\beta \in (0, 1)$ on the probability of a type II error is given (which may depend on the precise formulation of the alternative), we say the test has power $1 - \beta$.

**Maximum Mean Discrepancy.** The Maximum Mean Discrepancy (MMD) is a kernel-based measure of distance between two distributions $p$ and $q$, which is often used for two-sample testing. The MMD quantifies the dissimilarity between these distributions by comparing their mean embeddings in a reproducing kernel Hilbert space (RKHS; Aronszajn, 1950) with kernel $k(\cdot, \cdot)$. Formally, if $\mathcal{H}_k$ is the RKHS associated with kernel function $k$, the MMD between distributions $p$ and $q$ is the integral probability metric defined by:

$$\text{MMD}_k(p, q) \coloneqq \sup_{f \in \mathcal{H}_k : \|f\|_{\mathcal{H}_k} \leq 1} \left( \mathbb{E}_{X \sim p}[f(X)] - \mathbb{E}_{Y \sim q}[f(Y)] \right).$$

The minimum variance unbiased estimate of $\text{MMD}_k^2$, is given by the sum of two U-statistics and a sample average as:[2]

$$\widehat{\text{MMD}}_k^2(\boldsymbol{Z}) \coloneqq \frac{1}{n(n-1)} \sum_{(i,i') \in [n]_2} k(X_i, X_{i'}) + \frac{1}{m(m-1)} \sum_{(j,j') \in [m]_2} k(Y_j, Y_{j'}) - \frac{2}{mn} \sum_{i=1}^{n} \sum_{j=1}^{m} k(X_i, Y_j),$$

where we introduced the notation $[n]_2 = \{(i, i') \in [n]^2 : i \neq i'\}$ for the set of all pairs of distinct indices in $[n] = \{1, \ldots, n\}$. Tests based on the MMD usually reject the null when $\widehat{\text{MMD}}_k^2$ exceeds some critical threshold, with the resulting power being greatly affected by the kernel choice. For *characteristic* kernel functions (Sriperumbudur et al., 2011), it can be shown that $\text{MMD}_k(p, q) = 0$ if and only if $p = q$, leading to consistency results. However, on finite sample sizes, convergence rates of MMD estimates typically have strong dependence on the data dimension, so there are settings in which kernels ignoring redundant or unimportant features (*i.e.* non-characteristic kernels) will give higher test power in practice than characteristic kernels (which can over-weight redundant features).

**Distributions Over Kernels.** In our test statistic, we will consider the case where the kernel $k \in \mathcal{K}$ is drawn from a distribution $\rho \in \mathcal{M}_+^1(\mathcal{K})$ (with the latter notation denoting a probability measure; *c.f.* Benton et al., 2019 in the Gaussian Process literature). This distribution will be adaptively chosen based on the data subject to a regularisation term based on a "prior" $\pi \in \mathcal{M}_+^1(\mathcal{K})$ and defined through the Kullback-Liebler divergence: $\text{KL}(\rho\|\pi) \coloneqq \mathbb{E}_\rho[\log(\mathrm{d}\rho/\mathrm{d}\pi)]$ for $\rho \ll \pi$ and $\text{KL}(\rho\|\pi) \coloneqq \infty$ otherwise. When constructing these statistics the Donsker-Varadhan equality (Donsker and Varadhan, 1975), holding for any measurable $g : \mathcal{K} \to \mathbb{R}$, will be useful:

$$\sup_{\rho \in \mathcal{M}_+^1(\mathcal{A})} \mathbb{E}_\rho[g] - \text{KL}(\rho\|\pi) = \log \mathbb{E}_\pi[\exp \circ g]. \tag{1}$$

This can be further related to the notion of soft maxima. If $\pi$ is a uniform distribution on finite $\mathcal{K}$ with $|\mathcal{K}| = r$, then $\text{KL}(\rho\|\pi) \leq \log(r)$. Setting $g = tf$ for some $t > 0$, Equation (1) relaxes to

$$\max_k f(k) - \frac{\log(r)}{t} \leq \frac{1}{t} \log \left( \frac{1}{r} \sum_k e^{tf(k)} \right) \leq \max_k f(k), \tag{2}$$

---

[2]Kim et al. (2022) note that $\widehat{\text{MMD}}_k^2$ can equivalently be written as a two-sample U-statistic with kernel $h_k(x, x'; y, y') = k(x, x') + k(y, y') - k(x, y') - k(x', y)$, which is useful for analysis.

which approximates the maximum with error controlled by $t$. Our approach in considering these soft maxima is reminiscent of the PAC-Bayesian (McAllester, 1998; Seeger et al., 2001; Maurer, 2004; Catoni, 2007; see Guedj, 2019 or Alquier, 2021 for a survey) approach to capacity control and generalisation. This framework has raised particular attention recently as it has been used to provide the only non-vacuous generalisation bounds for deep neural networks (Dziugaite and Roy, 2017, 2018; Dziugaite et al., 2021; Zhou et al., 2019; Perez-Ortiz et al., 2021; Biggs and Guedj, 2021, 2022a); it has also been fruitfully applied to other varied problems from ensemble methods (Lacasse et al., 2006, 2010; Masegosa et al., 2020; Wu et al., 2021; Zantedeschi et al., 2021; Biggs and Guedj, 2022b; Biggs et al., 2022) to online learning (Haddouche and Guedj, 2022). Our proofs draw on techniques in that literature for U-Statistics (Lever et al., 2013) and martingales (Seldin et al., 2012; Biggs and Guedj, 2023; Haddouche and Guedj, 2023; Chugg et al., 2023). By considering these soft maxima, we gain a major advantage over the standard approaches, as we can obtain concentration and power results for our statistics without incurring Bonferroni-type multiple testing penalties.

**Permutation Tests.** The tests we discuss in this paper use permutations of the data $\boldsymbol{Z}$ to approximate the null distribution. We begin our discussion in a fairly general form to include other close settings such as independence testing (Albert et al., 2022; Rindt et al., 2021) or wild bootstrap-based two-sample tests (Fromont et al., 2012, 2013; Chwialkowski et al., 2014; Schrab et al., 2023). Let $\mathcal{G}$ be a group of transformations on $\boldsymbol{Z}$; in our setting $\mathcal{G} = \mathfrak{S}_{n+m}$, denoting the permutation (or symmetric) group of $[N]$ by $\mathfrak{S}_N$ and its elements by $\sigma \in \mathfrak{S}_N, \sigma : [N] \to [N]$. We write $g\boldsymbol{Z}$ for the action of $g \in \mathcal{G}$ on $\boldsymbol{Z}$; e.g. defining $\sigma\boldsymbol{Z} = (Z_{\sigma(1)}, \ldots, Z_{\sigma(n+m)})$ for the group elements $\sigma \in \mathfrak{S}_{n+m}$. We will suppose $\boldsymbol{Z}$ is invariant under the action of $\mathcal{G}$ when the sample is drawn from the *null* distribution, *i.e.* if the null is true than $g\boldsymbol{Z} =^d \boldsymbol{Z}$ (by which we notate equality of distribution) for all $g \in \mathcal{G}$. This is clearly the case for $\mathcal{G} = \mathfrak{S}_{n+m}$ in two-sample testing, since under the null $\boldsymbol{Z} = (Z_1, \ldots, Z_{m+n}) \overset{\text{iid}}{\sim} p$ with $p = q$, while under the alternative, the permuted sample $\sigma\boldsymbol{Z}$ for randomised $\sigma \sim \text{Uniform}(\mathfrak{S}_{n+m})$ simulates the null distribution.

We can use permutations to construct an approximate cumulative distribution function (CDF) of our test statistic under the null, and choose an appropriate quantile of this CDF as our test threshold, which must be exceeded under the null with probability less that level $\alpha$. For this we introduce a quantile operator (analogous to the max and min operators) for a finite set $\{f(a) \in \mathbb{R} : a \in \mathcal{A}\}$:

$$\underset{q,\,a\in\mathcal{A}}{\text{quantile}} f(a) := \inf \left\{ r \in \mathbb{R} : \frac{1}{|\mathcal{A}|} \sum_{a\in\mathcal{A}} \mathbf{1}\{f(a) \leq r\} \geq q \right\}. \tag{3}$$

Various different results can be used to choose the threshold giving a correct level; we will here highlight a very general and easy-to-use theorem for constructing permutation tests, and in Section 3 will discuss previously unconsidered implications for using unsupervised feature extraction methods as a part of our tests. Although this result is not new, we believe that its usefulness has been under-appreciated in the kernel testing community, and can be more conveniently applied than Romano and Wolf (2005, Lemma 1), which requires exchangeablity and is commonly used, *e.g.* in Albert et al. (2022, Proposition 1) and Schrab et al. (2023, Proposition 1).

**Theorem 1** (Hemerik and Goeman, 2018, Theorem 2.)**.** *Let $G$ be a vector of elements from $\mathcal{G}$, $G = (g_1, g_2, \ldots, g_{B+1})$, with $g_{B+1} = \text{id}$ (the identity permutation) for any $B \geq 1$. The elements $g_1, \ldots, g_B$ are drawn uniformly from $\mathcal{G}$ either i.i.d. or without replacement (which includes the possibility of $G = \mathcal{G}$). If $\tau(\boldsymbol{Z})$ is a statistic of $\boldsymbol{Z}$ and $\boldsymbol{Z} =^d g\boldsymbol{Z}$ for all $g \in \mathcal{G}$ under the null then*

$$\mathbb{P}_{p\times p,G} \left( \tau(\boldsymbol{Z}) > \underset{1-\alpha,\,g\in G}{\text{quantile}} \tau(g\boldsymbol{Z}) \right) \leq \alpha.$$

In other words, if we compare $\tau(\boldsymbol{Z})$ with the empirical quantile of $\tau(g\boldsymbol{Z})$ as a test threshold, the type I error rate is no more than $\alpha$.[3] The potentially complex task of constructing a permutation test reduces to the trivial task of choosing the statistic $\tau(\boldsymbol{Z})$. This result is also true for *randomised* permutations of any number $B \geq 1$ *without* approximation, so an exact and computationally efficient test can be straightforwardly constructed this way.

We finally mention the related approach of the MMD Aggregated test (MMDAgg; Schrab et al., 2023), which combines multiple MMD-based permutation tests with different kernels, and rejects if

---

[3]This test is near-exact. Specifically, if the statistic realisations are distinct ($\tau(g\boldsymbol{Z}) \neq \tau(g'\boldsymbol{Z})$ for $g \neq g'$) as is common for continuous data, the level is exactly $\lfloor (B+1)\alpha \rfloor / (B+1)$.

*any* of these reject, using distinct quantiles for each kernel. To ensure the overall aggregated test is well-calibrated, these quantiles must be adjusted using a *second level* of permutations. This incurs additional computational cost, a pitfall avoided by our fused single statistic.

**Faster Sub-Optimal Tests.** While the main focus of this paper revolves around the kernel selection problem for optimal, quadratic MMD testing, we also highlight the existence of a rich literature on efficient kernel tests which run in linear (or near-linear) time. These speed improvements are achieved using various tools such as: incomplete U-statistics (Gretton et al., 2012b; Yamada et al., 2019; Lim et al., 2020; Kübler et al., 2020; Schrab et al., 2022b), block U-statistics (Zaremba et al., 2013; Deka and Sutherland, 2022), eigenspectrum approximations (Gretton et al., 2009), Nyström approximations (Cherfaoui et al., 2022), random Fourier features (Zhao and Meng, 2015), analytic representations (Chwialkowski et al., 2015; Jitkrittum et al., 2016), deep linear kernels (Kirchler et al., 2020), kernel thinning (Dwivedi and Mackey, 2021; Domingo-Enrich et al., 2023), *etc.*

The efficiency of these tests usually entail weaker theoretical power guarantees compared to their quadratic-time counterparts, which are minimax optimal[4] (Kim et al., 2022; Li and Yuan, 2019; Fromont et al., 2013; Schrab et al., 2023; Chatterjee and Bhattacharya, 2023). These optimal quadratic tests are either permutation-based non-asymptotic tests (Kim et al., 2022; Schrab et al., 2023) or studentised asymptotic tests (Kim and Ramdas, 2023; Shekhar et al., 2022; Li and Yuan, 2019; Gao and Shao, 2022; Florian et al., 2023). We emphasise that the parameters of any of these permutation-based two-sample tests can be chosen in the unsupervised way we outline in Section 3. We choose to focus in this work on optimal quadratic-time results, but note that our general approach could be extended to sub-optimal faster tests as well.

## 3  Learning Statistics for Permutation Tests

Here we discuss how we can improve permutation-based tests by learning parameters for them in an unsupervised manner. The reason that we highlighted Theorem 1 in the previous section is that it holds for any $\tau$. For example, we could use the MMD with a kernel chosen based on the data, $\tau(\boldsymbol{Z}) = \widehat{\mathrm{MMD}}^2_{k=k(\boldsymbol{Z})}(\boldsymbol{Z})$; however, for each permutation $\sigma$ we would need to re-compute $\tau(\sigma\boldsymbol{Z}) = \widehat{\mathrm{MMD}}^2_{k=k(\sigma\boldsymbol{Z})}(\sigma\boldsymbol{Z})$, so the kernel being used would be different for each permutation. This has two major disadvantages: firstly, it might be computationally expensive to re-compute $k$ for each permutation, especially for a deep kernel[5]. Secondly, the *scale* of the resulting MMD could be dramatically different for each permutation, so the empirical quantile might not lead to a powerful test. This second problem is related to the problem of combining multiple different MMD values into a single test which our MMD-FUSE statistics are designed to combat (Section 4; *c.f.* also MMDAgg, Schrab et al., 2023).

**Our Proposal.** In two-sample testing we propose to use a statistic $\tau_\theta$ parameterised by some $\theta$, where $\theta$ is fixed for all permutations, but depends on the data in an *unsupervised* or *permutation-invariant* way. Specifically, we allow such a parameter to depend on $\langle \boldsymbol{Z} \rangle \coloneqq \{Z_1, \ldots, Z_{n+m}\}$, the *unordered* combined sample. Since our tests will not depend on the internal ordering of $\boldsymbol{X}$ and $\boldsymbol{Y}$ (which are assumed i.i.d. under both hypotheses), the only additional information contained in $\boldsymbol{Z}$ over $\langle \boldsymbol{Z} \rangle$ is the *label* assigning $Z_i$ to its initial sample. This is justified since $\langle \boldsymbol{Z} \rangle = \langle \sigma\boldsymbol{Z} \rangle$ for all $\sigma \in \mathfrak{S}_{n+m}$, so setting $\tau(\boldsymbol{Z}) = \tau_{\theta(\langle \boldsymbol{Z} \rangle)}(\boldsymbol{Z})$ gives a fixed $\theta$ and statistic for all permutations to use in Theorem 1. The information in $\langle \boldsymbol{Z} \rangle$ can be used to fine-tune test parameters for any test fitting this setup. This solves both the computation and scaling issues mentioned above.

The above provides a first formal justification for the use of the median heuristic in Gretton et al. (2012a), since it is a permutation-invariant function of the data. However, a far richer set of possibilities are available even when restricting ourselves to these permutation-free functions of $\boldsymbol{Z}$. For example, we can use any unsupervised or self-supervised learning method to learn representations to use as the input to an MMD-based test statistic, while paying no cost in terms of calibration and needing to train such methods only once. Given the wide variety of methods dedicated to feature extraction and dimensionality reduction, this opens up a huge range of possibilities for the design of new and principled two-sample tests. The simplicity and generality of our proposal might lead one to

---

[4]With the exception of the near-linear test of Domingo-Enrich et al. (2023) which achieves the same MMD separation rate as the quadratic test but under stronger assumptions on the data distributions.

[5]A possibility envisaged by *e.g.* Liu et al. (2020) and dismissed due to the computational infeasability.

expect that this idea has been used before, but it has not to our best knowledge, underlined by the fact that the median heuristic has been widely used without such justification when one follows from this method. This potentially powerful and widely-applicable possibility represents one of our most practical contributions.

# 4 MMD-FUSE: Fusing U-Statistics by Exponentiation

Say we have computed several MMD values $\widehat{\mathrm{MMD}}_k^2$ under different kernels $k \in \mathcal{K}$. How might we combine these? One possibility is to perform multiple testing as in the case of MMDAgg. An even simpler approach would simply take the maximum of those values $\max_k \widehat{\mathrm{MMD}}_k^2$ since Theorem 1 shows that this will not prevent us from controlling the level of our test by $\alpha$; note though that for each permutation we would take this maximum separately. Indeed, Cárcamo et al. (2022) show that for certain kernel choices the supremum of the MMD values with respect to the bandwidth is a valid integral probability metric (Müller, 1997). There are two main problems with this approach: a capacity control issue and a normalisation issue.

Firstly, if the class over which we are maximising is sufficiently rich (for example a complex neural network), then the maximum may be able to effectively memorise the entire sample for each possible permutation, saturating the statistic for every permutation and limiting test power. Any convergence results would need to hold *simultaneously* for every $k$ in $\mathcal{K}$, and so power results suffer: for finite $|\mathcal{K}|$ we would incur a sub-optimal Bonferroni correction (see Section 5); while for infinite classes, results would need to involve capacity control quantities like Rademacher complexity. Further, only information from a single maximising "base" kernel can be considered at a time.

Therefore, in both our statistics, we prefer a "soft" maximum, which considers information from every kernel simultaneously and when more than one of the kernels is well-suited (Section 5) it therefore avoids the Bonferroni correction arising from uniform bounds. From Equation (1), our approach is equivalent to using a KL complexity penalty, and is strongly reminiscent of *PAC-Bayesian* (Section 2) capacity control. We note that other soft maxima could be considered, but our choice makes obtaining exponential concentration inequalities relatively easy (Appendix C), and the dual formulation (Equation (2)) allows us to derive power results in terms of MMD directly.

The second issue is that the MMD estimates might have different scales or variances per kernel which need to be accounted for. In order to be able to meaningfully compare MMD values between each other, these need to be normalised somehow before taking a maximum. We use the common approach of dividing through by a variance-like term (as in "studentised" tests, see Section 2). The specific normaliser is permutation invariant and gives our statistic tight sub-Gaussian null concentration (for well-chosen regularisation parameter $\lambda$; see Appendix C).

Based on the above we introduce the FUSE-1 statistic which uses a log-sum-exp soft maximum, and the FUSE-N which combines this with normalisation. Both statistics use a "prior" distribution on $\mathcal{K}$, denoted $\pi(\langle \boldsymbol{Z} \rangle)$, which is either fixed independently of the data, or is a function of the data which is invariant under permutation (as discussed in Section 3).

**Definition 1.** *We define the un-normalised (subscript 1 for normaliser of 1) and normalised (subscript N) test statistics with parameter $\lambda > 0$, respectively, as*

$$\widehat{\mathrm{FUSE}}_1(\boldsymbol{Z}) := \frac{1}{\lambda} \log \left( \mathbb{E}_{k \sim \pi(\langle \boldsymbol{Z} \rangle)} \left[ \exp \left( \lambda \widehat{\mathrm{MMD}}_k^2(\boldsymbol{X}, \boldsymbol{Y}) \right) \right] \right),$$

$$\widehat{\mathrm{FUSE}}_N(\boldsymbol{Z}) := \frac{1}{\lambda} \log \left( \mathbb{E}_{k \sim \pi(\langle \boldsymbol{Z} \rangle)} \left[ \exp \left( \lambda \frac{\widehat{\mathrm{MMD}}_k^2(\boldsymbol{X}, \boldsymbol{Y})}{\sqrt{\widehat{N}_k(\boldsymbol{Z})}} \right) \right] \right),$$

*where $\widehat{N}_k(\boldsymbol{Z}) := \frac{1}{n(n-1)} \sum_{(i,j) \in [n+m]_2} k(Z_i, Z_j)^2$ is permutation invariant.*

Although these statistics appear complex, we note that in the case where $\pi$ has finite support, their calculation reduces to a log-sum-exp of MMD estimates normalised by $\widehat{N}_k$. This is even clearer when $\pi$ is also uniform on its support, as we consider experimentally; then Equation (2) shows that our statistics reduce to soft maxima, with $\lambda$ controlling the smoothness or "temperature".

From this, we define the FUSE-1 test (with the FUSE-N test $\Delta_{\text{FUSE}_N}$ defined analogously) as

$$\Delta_{\text{FUSE}_1}(\boldsymbol{Z}) := \mathbf{1}\left\{ \widehat{\text{FUSE}}_1(\boldsymbol{Z}) > \text{quantile } \widehat{\text{FUSE}}_1(\sigma \boldsymbol{Z}) \right\} \qquad (4)$$
$$\underset{1-\alpha, \sigma \in S}{}$$

for sampled set $S$ of permutations as described above. It compares the test statistic $\widehat{\text{FUSE}}_1$ with its quantiles under permutation, and rejects if the overall value exceeds a quantile controlled by $\alpha$. Note that since $\widehat{N}_k(\boldsymbol{Z})$ is permutation invariant, it only needs to be calculated once per kernel in $\Delta_{\text{FUSE}_N}$ (and not separately for each permutation) as per Section 3. See Appendix A.5 for its time complexity.

**Comparison with MMDAgg.** MMDAgg is a different way to think about combining multiple kernels and MMD values, but it relies on a framework based on multiple testing. This can be problematic in the case where the number of kernels considered is large, since as this number increases in the multiple testing approach MMDAgg, the level needs to be corrected differently, so its theretical power behaviour becomes unclear (though empirically MMDAgg retains its power in this setting). By contrast, our proposed FUSE methods avoid multiple testing as a single statistic and quantile are used, avoiding such issues. The FUSE statistics are even defined in the limit of an infinite number of kernels (continuous/uncountable collection of kernels) by considering distributions on the space of kernels, which is not the case for MMDAgg. Moreover, while the MMDAgg approach is only useful for hypothesis testing, having a quantity like FUSE combining multiple kernel-based measures of distance with exponential concentration bounds could be of interest in a wider range of applications.

**FUSE-1 and the Mean Kernel.** Although the un-normalised test has worse performance in practice than our normalised test, the FUSE-1 statistic is interesting in its own right because of various theoretical properties, as we discuss below. Firstly, we introduce the *mean* kernel $K_\rho(x, y) = \mathbb{E}_{k \sim \rho} k(x, y)$ under a "posterior" $\rho \in \mathcal{M}^1_+(\mathcal{K})$, which is indeed a reproducing kernel in its own right. In the finite case, this is simply a weighted sum of "base" kernels.

Note that the linearity of $\widehat{\text{MMD}}^2_k$ and $\text{MMD}^2_k$ in the kernel $k$ implies that $\mathbb{E}_{k \sim \rho} \text{MMD}^2_k(p, q) = \text{MMD}^2_{K_\rho}(p, q)$, and similarly for $\widehat{\text{MMD}}^2_k$, with these terms appearing in our power results (Section 5). Combining this linearity with the dual formulation of $\widehat{\text{FUSE}}_1$ via Equation (1) gives

$$\widehat{\text{FUSE}}_1(\boldsymbol{Z}) = \sup_{\rho \in \mathcal{M}^1_+(\mathcal{K})} \widehat{\text{MMD}}^2_{K_\rho}(\boldsymbol{X}, \boldsymbol{Y}) - \frac{\text{KL}(\rho, \pi)}{\lambda}.$$

This re-states $\widehat{\text{FUSE}}_1$ in terms of "posterior" $\rho$, and makes the interpretation of our statistic as a KL-regularised kernel-learning method clear. In the finite case, our test simply optimises the weightings of the different kernels in a constrained way. We note that for certain *infinite* kernel sets and choices of prior it is be possible to express the mean kernel in closed form. This happens because, *e.g.* the expectation of a Gaussian kernel with respect to a Gamma prior over the bandwidth is simply a (closed form) rational quadratic kernel. We discuss this point further in Appendix B.

## 5 Theoretical Power of MMD-FUSE

In this section we outline possible sufficient conditions for our tests to obtain power at least $1 - \beta$ at given level $\alpha$. The conditions will depend on a fixed data-independent "prior" $\pi \in \mathcal{M}^1_+(\mathcal{K})$ and thus hold even without unsupervised parameter optimisation. They are stated as requirements for the *existence* of a "posterior" $\rho \in \mathcal{M}^1_+(\mathcal{K})$ with corresponding mean kernel $K_\rho$ as defined in Section 4. In the finite case, $K_\rho$ is simply a weighted sum of kernels, so these requirements are also satisfied under the same conditions for any single kernel, corresponding to the case where the posterior puts all its weight on a single kernel.

Technically, these results require that there is some constant $\kappa$ upper bounding all of the kernels, and that $n$ and $m$ are within a constant multiple (*i.e.* $n \le m \le cn$ for some $c \ge 1$, notated $n \asymp m$). They hold when using randomised permutations provided $B > c' \alpha^{-2} \log(\beta^{-1})$ for small constant $c' > 0$.

**Theorem 2** (FUSE-1 Power). *Fix prior $\pi$ independently of the data. For the un-normalised test FUSE-1 with $\lambda \asymp n/\kappa$ and $n \asymp m$, there exists a universal constant $C > 0$ such that*

$$\exists \rho \in \mathcal{M}^1_+(\mathcal{K}) \ : \ \text{MMD}^2_{K_\rho}(p, q) > \frac{C\kappa}{n} \left( \frac{1}{\beta} + \log \frac{1}{\alpha} + \text{KL}(\rho, \pi) \right).$$

*is sufficient for FUSE-1 to achieve power at least $1 - \beta$ at level $\alpha$.*

A similar result is obtained for FUSE-N under an assumption that the normalising term is well behaved (see assumption in Theorem 3). This requirement will be satisfied for kernels (including most common ones) that tend to zero only in the limit of the data being infinitely far apart.

**Theorem 3** (FUSE-N Power). *Fix prior $\pi$ independently of the data such that for all $k \in \mathrm{supp}(\pi)$ the expectation $\mathbb{E}_{\boldsymbol{Z} \sim p \times q}[\widehat{N}_k(\boldsymbol{Z})^{-1}] < c/\kappa$ is bounded for some $c < \infty$. For the normalised test FUSE-N with $\lambda \asymp n \asymp m$, there exists a universal constant $C > 0$ such that*

$$\exists \rho \in \mathcal{M}_+^1(\mathcal{K}) \; : \; \mathrm{MMD}_{K_\rho}^2(p,q) > \frac{C\kappa}{n}\left(\frac{1}{\beta^2} + \log\frac{1}{\alpha} + \mathrm{KL}(\rho,\pi)\right).$$

*is sufficient for FUSE-N to achieve power at least $1 - \beta$ at level $\alpha$.*

**Discussion.** The conditions in Theorems 2 and 3 give the optimal $\mathrm{MMD}^2$ separation rate in $n$ (Domingo-Enrich et al., 2023, Proposition 2). These results also imply consistency if the prior $\pi$ assigns non-zero weight to characteristic kernels.[6] Applying these results to uniform priors supported on $r$ points, the KL penalty can be upper bounded as $\mathrm{KL}(\rho,\pi) \leq \log(r)$. Thus in the *worst* case, where only a single kernel achieves large $\mathrm{MMD}_k^2(p,q)$, the price paid for adaptivity is only $\log(r)$. In many cases, *most* of the kernels will give large $\mathrm{MMD}_k^2(p,q)$. The posterior will then mirror the prior, and this KL penalty will be even smaller. Thus very large numbers of kernels could be considered, and if all give large MMD values the power would not be greatly affected.

**Additional Technical Results.** Aside from the presentation of our new statistics and tests, we make a number of technical contributions on the way to proving Theorems 2 and 3, as well as proving some additional results. In particular, we give exponential concentration bounds for our statistics under permutations and the null, which do not require bounded kernels. This refined analysis requires the construction of a coupled Rademacher chaos and concentration thereof. We obtain intermediate results using variances from the proofs of Theorems 2 and 3 that could be used in future work to obtain power guarantees under alternative assumptions such as Sobolev spaces (Schrab et al., 2023). Finally, we prove exponential concentration for $\widehat{\mathrm{FUSE}}_1$ under the alternative and bounded kernels, requiring the proof of a "PAC-Bayesian" bounded differences-type concentration inequality. See the appendix for a more detailed overview.

## 6 Experiments

We compare the test power of MMD-FUSE-N ($\lambda = \sqrt{n(n-1)}$) against various MMD-based kernel selective tests (see Section 1 for details) using: the median heuristic (MMD-Median; Gretton et al., 2012a), data splitting (MMD-Split; Sutherland et al., 2017), analytic Mean Embeddings and Smooth Characteristic Functions (ME & SCF; Jitkrittum et al., 2016), the MMD Deep kernel (MMD-D; Liu et al., 2020), Automated Machine Learning (AutoML; Kübler et al., 2022b), kernel thinning to (Aggregate) Compress Then Test (CTT & ACTT; Domingo-Enrich et al., 2023), and MMD Aggregated (Incomplete) tests (MMDAgg & MMDAggInc; Schrab et al., 2023, 2022b). Additional details and code link for experimental reproducibility are provided in Appendix A.

**Distribution on Kernels.** We choose our kernel prior distribution $\pi$ as uniform over a collection of Gaussian, $k_\gamma^g(x,y) = \exp\left(-\|x-y\|_2^2/2\gamma^2\right)$, and Laplace, $k_\gamma^\ell(x,y) = \exp\left(-\sqrt{2}\|x-y\|_1/\gamma\right)$, kernels with various bandwidths $\gamma > 0$. These bandwidths are chosen as the uniform discretisation of the interval between half the 5% and twice the 95% quantiles (for robustness) of $\{\|z - z'\|_r : z, z' \in \boldsymbol{Z}\}$, with $r \in \{1, 2\}$, respectively. This choice is similar to that of Schrab et al. (2023, Section 5.2), who empirically show that ten points for the discretisation is sufficient (Schrab et al., 2023, Figure 6), which we verify in Appendix A.4. This set of distances is permutation-invariant, so Theorem 1 guarantees a well-calibrated test even though the kernels are data-dependent.

**Mixture of Gaussians.** Our first experiments (Figure 1) consider multimodal distributions $p$ and $q$, each a 2-dimensional mixture of four Gaussians with means $(\pm\mu, \pm\mu)$ with $\mu = 20$ and diagonal covariances. For $p$, the four components all have unit variances, while for $q$ we vary the standard deviation $\sigma$ of *one* of the Gaussians, $\sigma = 1$ corresponds to the null hypothesis $p = q$. Intuitively, an appropriate kernel bandwidth to distinguish $p$ from $q$ would correspond to that separating Gaussians

---

[6]Under this condition $\mathrm{KL}(\nu, \pi) < \infty$ with $\nu$ the restriction of $\pi$ to characteristic kernels. $K_\nu$ is characteristic so $\mathrm{MMD}_{K_\nu}(p,q) > 0 \iff p \neq q$, and our condition lower bound tends to zero with $n \to \infty$.

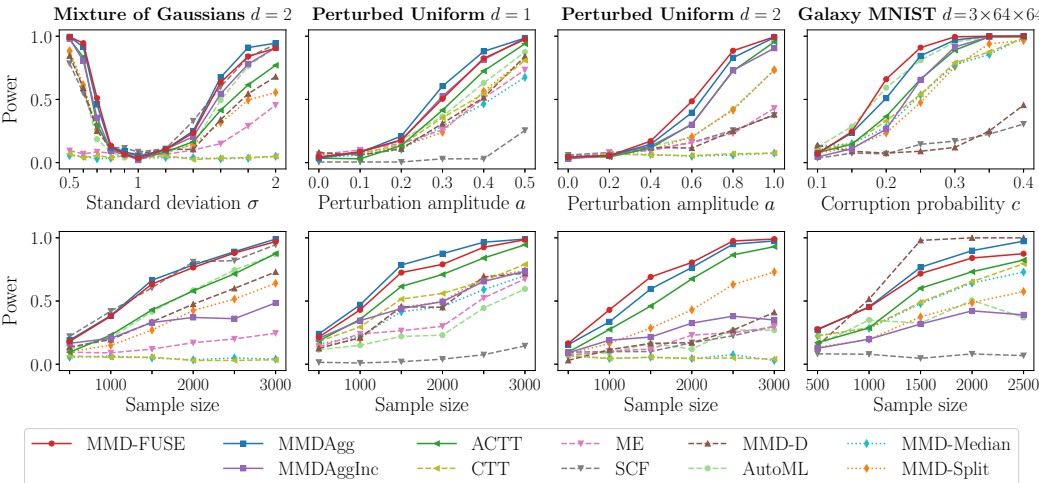

Figure 1: Power experiments. The four columns correspond to different settings: Mixture of Gaussians in two dimensions (null $\sigma = 1$), Perturbed Uniform for $d \in \{1, 2\}$ (null $a = 0$), and Galaxy MNIST in dimension 12288 (null $c = 0$). In the first row, the deviations away from the null are varied for fixed sample size $m = n = 500$. In the second row, the sample size varies while the deviations are fixed as $\sigma = 1.3$, $a = 0.2$, $a = 0.4$, and $c = 0.15$, for the four respective problems. The plots correspond to the rejections of the null averaged over 200 repetitions.

with standard deviations 1 and $\sigma$. This is significantly smaller than the *median* bandwidth which scales with the distance $\mu$ between modes.

**Perturbed Uniform.** In Figure 1, we report test power for detecting perturbations on uniform distributions in one and two dimensions. We vary the amplitude $a$ of two perturbations from $a = 0$ (null) to $a = 1$ (maximum value for the density to remain non-negative). A similar benchmark was first proposed by Schrab et al. (2023, Section 5.5) and considered in several other works (Schrab et al., 2022b; Hagrass et al., 2022; Chatterjee and Bhattacharya, 2023). Different bandwidths are required to detect different amplitudes of the perturbations.

**Galaxy MNIST.** We examine performance on real-world data in Figure 1, through galaxy images (Walmsley et al., 2022) in dimension $d = 3 \times 64 \times 64 = 12288$ captured by a ground-based telescope. These consist of four classes: 'smooth and cigar-shaped', 'edge-on-disk', 'unbarred spiral', and 'smooth and round'. One distribution uniformly samples images from the first three categories, while the other does the same with probability $1 - c$ and uniformly samples a 'smooth and round' galaxy image with probability of corruption $c \in [0, 1]$. The null hypothesis corresponds to the case $c = 0$.

**CIFAR 10 vs 10.1.** The aim of this experiment is to detect the difference between images from the CIFAR-10 (Krizhevsky, 2009) and CIFAR-10.1 (Recht et al., 2019) test sets. This is a challenging problem as CIFAR-10.1 was specifically created to consist of new samples from the CIFAR-10 distribution so that it can be used as an alternative test set for models trained on CIFAR-10. Samples from the two distributions are presented in Figure 6 in Appendix A.3 (Liu et al., 2020, Figure 5). This

Table 1: Test power for detecting the difference between CIFAR-10 and CIFAR-10.1 images with test level $\alpha = 0.05$. The averaged numbers of rejections over 1000 repetitions are reported.

| Tests | Power |
|---|---|
| MMD-FUSE | **0.937** |
| MMDAgg | 0.883 |
| MMD-D | 0.744 |
| CTT | 0.711 |
| MMD-Median | 0.678 |
| ACTT | 0.652 |
| ME | 0.588 |
| AutoML | 0.544 |
| C2ST-L | 0.529 |
| C2ST-S | 0.452 |
| MMD-O | 0.316 |
| MMDAggInc | 0.281 |
| SCF | 0.171 |

benchmark was proposed by Liu et al. (2020, Table 3) who introduced the deep MMD test MMD-D and the MMD-Split test (here referred to as MMD-O to point out that their implementation has been used rather than ours). They also compare to ME and SCF, as well as to C2ST-L and C2ST-S (Lopez-Paz and Oquab, 2017) which correspond to Classifier Two-Sample Tests based on Sign or

Linear kernels. For the tests slitting the data, 1000 images from both datasets are used for parameter selection and/or model training, and 1021 other images from each distributions are used for testing. Consequently, tests avoiding data splitting are given the full 2021 images from CIFAR-10.1 and 2021 images sampled from CIFAR-10.

**Experimental Results of Figure 1.** We observe similar trends in all eight experiments in Figure 1: MMD-FUSE *matches the power* of state-of-the-art MMDAgg while being *computationally faster* (both theoretically and practically). These two tests consistently obtain the highest power in every experiment, except when increasing the number of Galaxy MNIST images where MMD-D obtains higher power. However, we observe in Table 1 that MMD-FUSE outperforms MMD-D on another image data problem, also with large sample size. On synthetic data, the deep kernel test MMD-D surprisingly only obtains power similar to MMD-Split in most experiments (even lower in the 2-dimensional perturbed uniform experiment). The two near-linear aggregated variants ACTT and MMDAggInc trade-off a small portion of test power for computational efficiency, with the former outperforming the latter for large sample sizes. The importance of kernel selection is emphasised by the fact that the two tests using the median bandwidth (MMD-Median and CTT) achieve very low power. While the linear-time SCF test, based in the frequency domain, attains high power in the Mixture of Gaussians experiments, it has low power in the three other experiments. Its spatial domain variant ME performs better on Perturbed Uniform $d \in \{1, 2\}$ experiments but in general still has reduced power compared to both linear and quadratic time alternatives. Finally, the AutoML test performs well for fixed sample size $m = n = 500$ (first row of Figure 1), but its power compared to other tests considerably deteriorates as the sample size increases (second row of Figure 1). Overall, MMD-FUSE achieves *state-of-the-art performance* across all experiments on both low-dimensional synthetic data and high-dimensional real-world data.

**Experimental Results of Table 1.** We report in Table 1 the power achieved by each test on the CIFAR 10 vs 10.1 experiment, which is averaged over 1000 repetitions. We observe that MMD-FUSE performs the best and obtains power 0.937, which means that out of 1000 repetitions, it was 937 times able to distinguish between samples from CIFAR-10 and from CIFAR-10.1. This demonstrates that the images in CIFAR-10.1 do not come from the same distribution as those in CIFAR-10.

**Experimental Results of Figure 7.** We observe in Figure 7 of Appendix A.4 that MMD-FUSE can achieve higher power than MMDAgg in some additional perturbed uniform experiment. We also note that using a relatively small number of kernels (*e.g.*, 10) is enough to capture all the required information, and that the test power is retained when further increasing the number of kernels.

## 7 Conclusions

In this work, we propose MMD-FUSE, an MMD-based test which fuses kernels through a soft maximum and a method for learning general two-sample testing parameters in an unsupervised fashion. We demonstrate the empirical performance of MMD-FUSE and show that it achieves the optimal MMD separation rate guaranteeing high test power. This optimality holds with respect to the sample size and likely also for the logarithmic dependence in $\alpha$, but we believe the dependence on $\beta$ could be improved in future work; a general question is whether lower bounds in terms of $\alpha$ and $\beta$ can be proved. Obtaining separation rates in terms of the $L^2$-norm between the densities (Schrab et al., 2023) may also be possible but challenging since this distance is independent of the kernel.

An open question is in explaining the significant empirical power advantage of the normalised test over its un-normalised variant, which is currently not reflected in the derived rates. The importance of this normalisation is clear when considering kernels with different bandwidths, leading to vastly different scaling in the un-normalised permutation distributions. Work here could begin with finite-sample concentration guarantees for our normalised statistic or other "studentised" variants, some of which might obtain better performance.

Future work could also examine computationally efficient variants of MMD-FUSE, by either relying on incomplete $U$-statistics (Schrab et al., 2022b) and leading to suboptimal rates, or by relying on recent ideas of kernel thinning (Domingo-Enrich et al., 2023) which can lead to the same optimal rate under stronger assumptions on the data distributions, or by considering the permutation-free approach of Shekhar et al. (2022). Finally, our two-sample MMD fusing approach could be extended to the HSIC independence framework (Gretton et al., 2005, 2008; Albert et al., 2022) and to the KSD goodness-of-fit setting (Chwialkowski et al., 2016; Liu et al., 2016; Schrab et al., 2022a).

**Acknowledgements**

We would like to thank the anonymous reviewers and area chair for their thorough reading of our work, for their constructive feedback, and for engaging in the discussions, all of which have helped to improve the paper. Felix Biggs and Antonin Schrab both gratefully acknowledge the support from the U.K. Research and Innovation under the EPSRC grant EP/S021566/1. Arthur Gretton acknowledges support from the Gatsby Charitable Foundation.

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

## Overview of Appendices

**Appendix A** We give further experimental details, discuss time complexity of our results while graphing run-times.

**Appendix B** We discuss how $\widehat{\mathrm{FUSE}}_1$ can be expressed in a simple form for certain uncountable priors.

**Appendix C** We prove exponential concentration bounds for both our statistics under the null and permutations.

**Appendix D** We prove exponential concentration bounds for the $\widehat{\mathrm{FUSE}}_1$ statistic under the alternative hypothesis.

**Appendix E** We prove the power results stated in Section 5 and other interesting intermediate results.

## A    Additional Experimental Details

### A.1    Code and licenses

Our implementation of MMD-FUSE in Jax (Bradbury et al., 2018), as well as the code for the reproducibility of the experiments, are made publicly available at:



https://github.com/antoninschrab/mmdfuse-paper



Our code is under the MIT License, and we also implement ourselves the MMD-Median (Gretton et al., 2012a) and MMD-Split (Sutherland et al., 2017) tests. For ME & SCF (Jitkrittum et al., 2016), MMD-D (Liu et al., 2020), AutoML (Kübler et al., 2022b), CTT & ACTT (Domingo-Enrich et al., 2023), MMDAgg & MMDAggInc (Schrab et al., 2023, 2022b), we use the implementations of the respective authors, which are all under the MIT license.

The experiments were run on an AMD Ryzen Threadripper 3960X 24 Cores 128Gb RAM CPU at 3.8GHz and on an NVIDIA RTX A5000 24Gb Graphics Card, with a compute time of a couple of hours.

### A.2    Test parameters

In general, we use the default test parameters recommended by the authors of the tests (listed above). For the ME and SCF tests, ten test locations are chosen on half of the data. AutoML is run with the recommended training time limit of one minute.

**Kernels.** As explained in Section 6 (§Distribution on Kernels), for MMD-Fuse, we use Gaussian and Laplace kernels with bandwidths in $\{q_{5\%}^r + i(q_{95\%}^r - q_{5\%}^r)/9 : i = 0, \ldots, 9\}$ where $q_{5\%}^r$ is half the 5% quantile of all the inter-sample distances $\{\|z - z'\|_r : z, z' \in \mathbf{Z}\}$ with $r = 1$ and $r = 2$ for Laplace and Gaussian kernels, respectively. Similarly, $q_{95\%}^r$ is twice the 95% quantile.

MMD-Split selects a Gaussian kernel on half of the data with bandwidth in $\{q_{5\%}^r + i(q_{95\%}^r - q_{5\%}^r)/99 : i = 0, \ldots, 99\}$ by maximizing a proxy for asymptotic power which is the ratio of the estimated MMD with its estimated standard deviation under the alternative (Liu et al., 2020, Equation 3). The MMD test is then run on the other half with the selected kernel.

CTT and MMD-Median both use a Gaussian kernel with bandwidth the median of $\{\|z - z'\|_2 : z, z' \in \mathbf{Z}\}$, while ACTT is run with Gaussian kernels with bandwidths in $\{q_{5\%}^2 + i(q_{95\%}^2 - q_{5\%}^2)/9 : i = 0, \ldots, 9\}$.

The MMDAgg and MMDAggInc tests are run with their default implementations, which similarly use collections of 20 kernels split equally between Gaussian and Laplace kernels with ten bandwidths each, but they use a different (non-uniform) discretisation of the intervals $[q_{95\%}^r, q_{5\%}^r]$.

**Permutations.** For MMD-Median, MMD-Split, and MMD-FUSE, we use 2000 permutations to estimate the quantiles. MMDAgg and MMDAggInc use $2000 + 2000$ and $500 + 500$ permutations, respectively, to approximate the quantiles and the multiple testing correction. The CTT and ACTT tests are run with 39 and $299 + 200$ permutations, respectively. AutoML uses 10000 permutations

and MMD-D 100 of them. The ME and SCF tests use asymptotic quantiles. We recall that using a higher number of permutations for the quantile does not necessarily lead to a more powerful test (Rindt et al., 2021, Section 7).

## A.3 Details on the experiments of Section 6

In this section, we present figures illustrating the four experimental settings described in Section 6 with results in Figure 1: Mixture of Gaussians (Figure 2), Perturbed Uniform $d = 1$ (Figure 3), Perturbed Uniform $d = 2$ (Figure 4), Galaxy MNIST (Figure 5), and CIFAR 10 vs 10.1 (Figure 6).

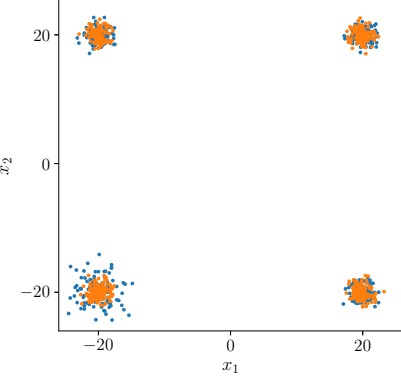

Figure 2: Mixture of Gaussians. Both distributions are mixtures of four 2-dimensional Gaussians with means $(\pm 20, \pm 20)$ and standard deviations $\sigma_1 = \sigma_2 = \sigma_3 = 1$. Samples from $p$ *(orange)* are drawn with $\sigma_4 = 1$, while samples from $q$ *(blue)* are drawn with $\sigma_4 = 2$.

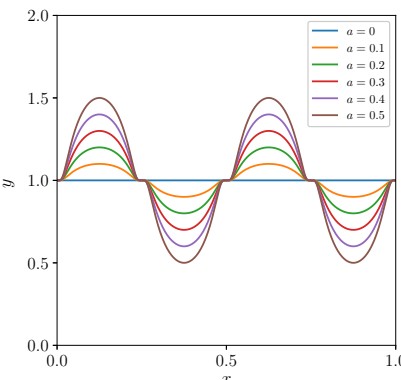

Figure 3: Perturbed Uniform $d = 1$. One-dimensional uniform densities with two perturbations are plotted for various amplitudes $a$.

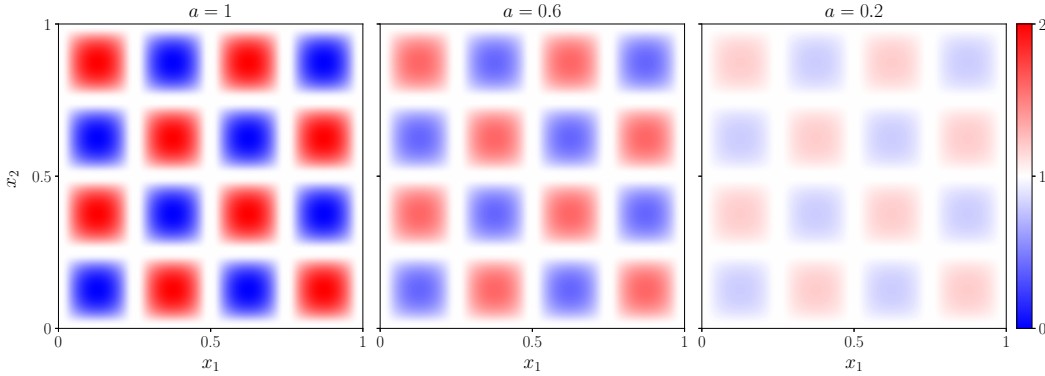

Figure 4: Perturbed Uniform $d = 2$. Two-dimensional uniform densities with two perturbations per dimension are plotted for various amplitudes $a$.

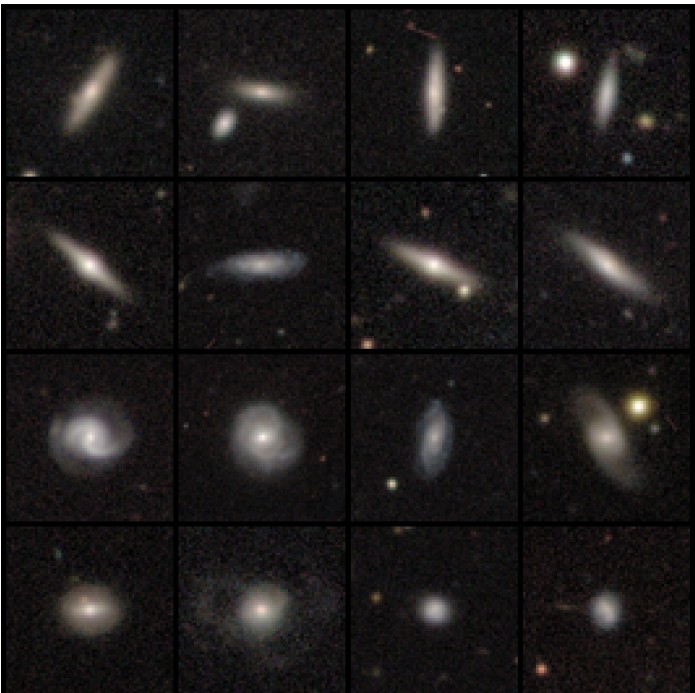

Figure 5: Galaxy MNIST (Walmsley et al., 2022) images in dimension $3 \times 64 \times 64$ across four categories: 'smooth cigar' *(first row)*, 'edge on disk' *(second row)*, 'unbarred spiral' *(third row)*, and 'smooth round' *(fourth row)*.

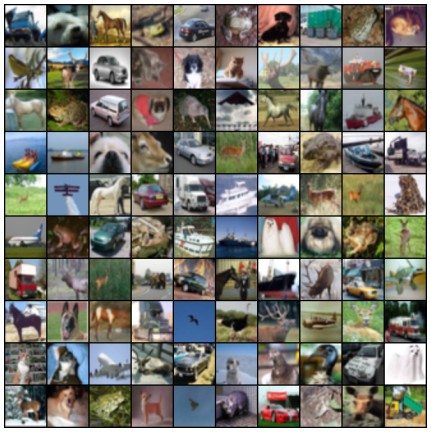

(a) CIFAR-10 images

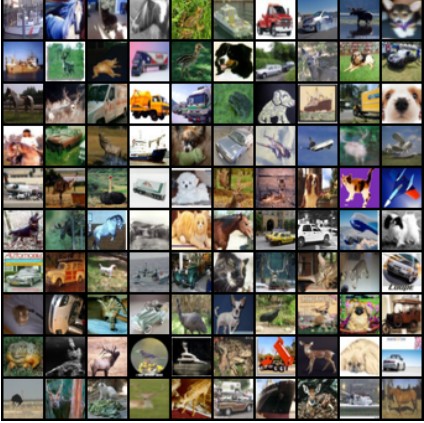

(b) CIFAR-10.1 images

Figure 6: Images from the CIFAR-10 (Krizhevsky, 2009) and CIFAR-10.1 (Recht et al., 2019) test sets. This figure corresponds to Figure 5 of Liu et al. (2020).

## A.4 Experiment varying the number of kernels

In Figure 7, we provide an additional perturbed uniform experiment where we vary the number of kernels for MMD-FUSE and for MMDAgg. We consider the task of detecting six one-dimensional perturbations with amplitude $a = 0.5$ for sample sizes $m = n = 500$ while varying the number of Gaussian and Laplace kernels from 10 to 1000.

Firstly, we observe that MMD-FUSE achieves higher power than MMDAgg in this setting. Secondly, both MMD-FUSE and MMDAgg retain their power when increasing the number of bandwidths. This matches the empirical observation of Schrab et al. (2023, Section 5.7) that a discretisation of 10 bandwidths is enough to capture all the information in certain two-sample problems and that using a finer discretisation (which is computationally more expensive) does not improve the test power.

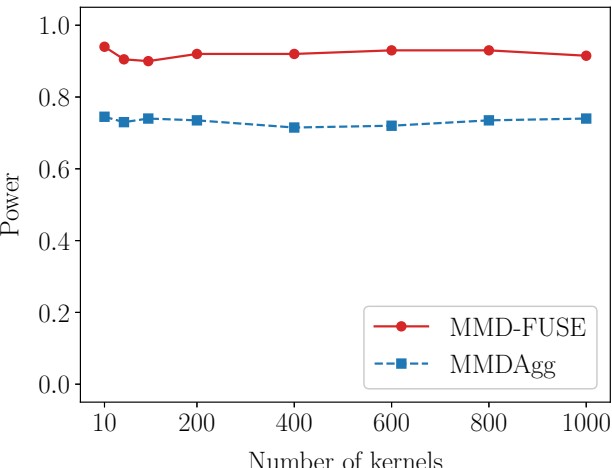

Figure 7: Power experiment with perturbed uniform samples while varying the number of kernels.

## A.5 Time complexity and runtimes

The time complexity of MMD-FUSE is

$$\mathcal{O}\Big( K\, B\, \big(m^2 + n^2\big)\Big) \tag{5}$$

where $m$ and $n$ are the sizes of the two samples, $K$ is the number of kernels fused, and $B$ is the number of permutations to estimate the quantile. Note that this is an improvement over the time complexity of MMDAgg (Schrab et al., 2023)

$$\mathcal{O}\Big( K\, \big(B + B'\big)\big(m^2 + n^2\big)\Big) \tag{6}$$

where the extra parameter $B'$ corresponds to the number of permutations used to estimate the multiple testing correction, often set as $B' = B$ in practice (Schrab et al., 2023, Section 5.2). We indeed observe in Figure 8 that MMD-FUSE runs twice as fast as MMDAgg.[7]

While the runtimes of most tests should not depend on the type of data (only on the sample size and on the dimension), we note that the runtimes of tests relying on optimisation (*e.g.* ME & SCF) can be affected. In the experiment of Figure 8, we consider samples from multivariate Gaussians centred at zero with covariance matrices $I_d$ and $\sigma I_d$ with $\sigma = 1.1$. We vary both the sample sizes and the dimensions.

---

[7]The two constants hidden in the $\mathcal{O}$-notations of Equations (5) and (6) are exactly the same, which is indeed verified by our empirical observations.

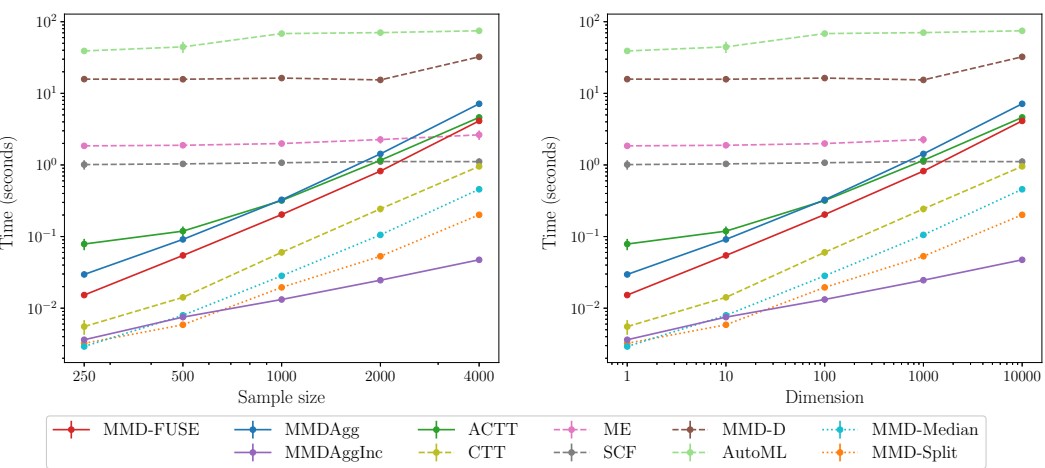

Figure 8: Test runtimes plotted using logarithmic scales. In the LHS figure, we vary the sample size for fixed dimension 10. In the RHS figure, we vary the dimension for fixed sample size 500. The mean and standard deviations of the test runtimes over ten repetitions are reported. Recall that the tests are run with their default parameters with different numbers of permutations, and that AutoML has a training time parameter which is set to 60 seconds by default, as described in Appendix A.2.

# B    Closed Form Mean Kernel

We have observed in the main text that in some cases where the prior has non-finite support, $\widehat{\mathrm{FUSE}}_1$ still gives a straightforwardly expressed statistic. This happens because for certain kernel choices, the expectation of the kernel with respect to some parameter is still a closed form kernel. We give one example of such a statistic here.

**Theorem 4.** *For any fixed $\alpha > 0$, the following is an example of a $\widehat{\mathrm{FUSE}}_1$ statistic:*

$$\widehat{\mathrm{FUSE}}_1^{rq}(\boldsymbol{Z}) = \sup_{R>0} \widehat{\mathrm{MMD}}_{k_{rq(\alpha,\sqrt{R}\eta_0)}}^2(\boldsymbol{Z}) - \alpha \cdot \frac{\log R + 1/R - 1}{\lambda}$$

*where $k_{rq(\alpha,\eta)} = (1 + \|x - y\|^2/2\eta^2)^{-\alpha}$ is a rational quadratic kernel, and $\eta_0$ is the "prior" bandwidth (which can essentially be absorbed into the data scaling).*

**Proof of Theorem 4.** Firstly, we define a general rational quadratic kernel

$$k_{rq(\alpha,\eta)}(x,y) = \left(1 + \frac{\|x - y\|_2^2}{2\eta^2}\right)^{-\alpha}.$$

Note that $k_g(r) = e^{-\tau r^2/2}$ with $r = \|x - y\|_2$ is a bounded kernel with parameter $\tau$. If we take the expectation of $\tau$ with respect to a Gamma distribution,

$$\mathbb{E}_{\tau \sim \Gamma(\alpha,\beta)} e^{-\tau r^2/2} = \frac{\beta^\alpha}{\Gamma(\alpha)} \int_{-\infty}^\infty \tau^{\alpha-1} \exp(-\tau\beta) \exp(-\tau r^2/2)\, \mathrm{d}\tau$$

$$= \frac{\beta^\alpha}{\Gamma(\alpha)} \Gamma(\alpha) \left(\beta + \frac{r^2}{2}\right)^{-\alpha}$$

$$= \left(1 + \frac{r^2}{2\beta}\right)^{-\alpha} = k_{rq(\alpha,\sqrt{\beta})}(r)$$

where $\Gamma$ denotes the gamma function. The KL divergence between Gamma distributions is

$$\mathrm{KL}(\Gamma(\alpha,\beta),\Gamma(\alpha_0,\beta_0)) = (\alpha - \alpha_0)\psi(\alpha) + \log\frac{\Gamma(\alpha_0)}{\Gamma(\alpha)} + \alpha_0 \log R + \alpha\left(\frac{1}{R} - 1\right)$$

where $R = \beta/\beta_0$ and $\psi(\alpha) := \Gamma'(\alpha)/\Gamma(\alpha)$ denotes the digamma function. Using the dual form of $\widehat{\mathrm{FUSE}}_1$ we find that

$$\widehat{\mathrm{FUSE}}_1 = \sup_{\alpha>0,\beta>0} \widehat{\mathrm{MMD}}_{k_{rq(\alpha,\sqrt{\beta})}}^2(\boldsymbol{Z}) - \frac{\mathrm{KL}(\Gamma(\alpha,\beta),\Gamma(\alpha_0,\beta_0))}{\lambda}.$$

Restricting $\alpha = \alpha_0$, prior $\beta_0 = \eta_0^2$ and setting $\beta = R\eta_0^2$ gives the result. $\qquad\square$

# C  Null and Permutation Concentration Results

In this section we prove the following concentration results under the null (or equivalently, under the permutation distribution). These results show that for appropriate choices of $\lambda$, both statistics converge to zero at rate at least $\mathcal{O}(1/n)$ under the null or permutations. We recall that without loss of generality $n \leq m$.

**Theorem 5.** *For bounded kernels $k \leq \kappa < \infty$ and parameter $\lambda$, with probability at least $1 - \delta$ over a sample $\boldsymbol{Z}$ from the null,*

$$\widehat{\mathrm{FUSE}}_1(\boldsymbol{Z}) \leq \frac{4\kappa^2\lambda}{n(n-1)} + \frac{\log\frac{1}{\delta}}{\lambda}$$

*provided $0 < \lambda < \sqrt{n(n-1)}/8\sqrt{2}\kappa$, and*

$$-\widehat{\mathrm{FUSE}}_1(\boldsymbol{Z}) \leq \frac{1 + 32\log\frac{1}{\delta}}{\sqrt{2n(n-1)}} \cdot \frac{\kappa}{2}.$$

*For potentially unbounded kernels and parameter $\lambda$, with probability at least $1 - \delta$ over a sample $\boldsymbol{Z}$ from the null,*

$$\widehat{\mathrm{FUSE}}_N(\boldsymbol{Z}) \leq \frac{16\lambda}{n(n-1)} + \frac{\log\frac{1}{\delta}}{\lambda},$$

*provided $0 < \lambda < \sqrt{n(n-1)}/16\sqrt{2}$, and*

$$-\widehat{\mathrm{FUSE}}_N(\boldsymbol{Z}) \leq \frac{1 + 32\log\frac{1}{\delta}}{\sqrt{2n(n-1)}}.$$

*The above bounds are also valid for $\widehat{\mathrm{FUSE}}_1(\sigma\boldsymbol{Z}), \widehat{\mathrm{FUSE}}_N(\sigma\boldsymbol{Z})$ and any fixed $\boldsymbol{Z}$ (potentially non-null) under permutation by $\sigma \sim \mathrm{Uniform}(\mathfrak{S}_{n+m})$.*

While the upper bounds depend critically upon our choice of $\lambda$, the lower bounds instead hold regardless of $\lambda$. Choosing $\lambda \asymp \sqrt{n(n-1)} \asymp n$ gives the desired upper bound rates $\mathcal{O}(1/n)$.

## C.1  Sub-Gaussian Chaos Theorem

We provided constants for Theorem 5, even though they are not strictly necessary, as we hope they could be useful when adapting FUSE statistics to other settings. In particular, the constants help understand for which values of $\lambda$ the bounds hold, and to understand the relation between the two different right hand side terms which can be matched by tuning $\lambda$. In provide these constants, in this subsection we replicate the proof of a sub-result from Rudelson and Vershynin (2013), but unlike in that paper we explicitly keep track of the constants. To be clear, Rudelson and Vershynin (2013) does not give numerical constants at all (only their existence is proved), and we need then for our statement of Theorem 5. We do not claim this as a contribution, the proof is only included so that we are not stating the numerical constants without justification.

**Theorem 6** (Sub-Gaussian Chaos; adapted from Rudelson and Vershynin, 2013)**.** *Let $X_i$ be mean-zero 1-sub-Gaussian variables, such that $\log \mathbb{E}_X \exp(tX) \leq \frac{1}{2}t^2$ for every $t \in \mathbb{R}$. Let $A \in \mathbb{R}^{n \times n}$ be a real symmetric matrix with zeros on the diagonal and we define the sub-Gaussian chaos*

$$W := \sum_{i=1}^n \sum_{j=1}^n A_{ij} X_i X_j.$$

*For all $|t| \leq (4\sqrt{2}\|A\|)^{-1}$,*

$$\mathbb{E}_X \exp(tW) \leq \exp\left(16t^2\|A\|_F^2\right).$$

**Proof.** This proof is closely adapted from Rudelson and Vershynin (2013) with some modifications to explicitly track numerical constants.

Let $X_i$ be mean-zero 1-sub-Gaussian variables, such that $\log \mathbb{E}_X \exp(tX) \leq \frac{1}{2}t^2$ for every $t \in \mathbb{R}$. We are considering

$$W := \sum_{i,j} A_{ij} X_i X_j$$

where $A_{ij}$ has zeros on the diagonal. We introduce independent Bernoulli random variables $\delta_i \in \{0, 1\}$ with $\mathbb{E}\delta_i = 1/2$ and define the matrix $A^\delta$ with entries $(A^\delta)_{ij} = \delta_i(1 - \delta_j)A_{ij}$. Note that $\mathbb{E}_\delta A^\delta = \frac{1}{4}A$, since $\mathbb{E}\delta_i(1 - \delta_j)$ equals $1/4$ for $i \neq j$.

Writing the set of indices $\Lambda_\delta = \{i \in [n] : \delta_i = 1\}$, we introduce

$$W_\delta := \sum_{i,j} A^\delta_{ij} X_i X_j = \sum_{i \in \Lambda_\delta, j \in \Lambda_\delta^c} A_{ij} X_i X_j = \sum_{j \in \Lambda_\delta^c} X_j \Big( \sum_{i \in \Lambda_\delta} A_{ij} X_i \Big).$$

By Jensen's inequality,
$$\mathbb{E}_X \exp(tW) \leq \mathbb{E}_{X,\delta} \exp(4tW_\delta).$$

Conditioned on $\delta$ and $(X_i)_{i \in \Lambda_\delta}$, $W_\delta$ is a linear combination of the mean-zero sub-gaussian random variables $X_j$, $j \in \Lambda_\delta^c$. It follows that this conditional distribution of $W_\delta$ is sub-gaussian, and so

$$\mathbb{E}_{(X_j)_{j \in \Lambda_\delta^c}} \exp(4tW_\delta) \leq \exp\left( \frac{1}{2}(4t)^2 \sum_{j \in \Lambda_\delta^c} \Big( \sum_{i \in \Lambda_\delta} A_{ij} X_i \Big)^2 \right).$$

Now introduce $g = (g_1, \ldots, g_n) \sim \mathcal{N}(0, 1)$ i.i.d. draws from a normal and note that the above can also arise directly as the moment generating function of these Normal variables, so we make the further equality (for fixed $X$ and $\delta$),

$$\exp\left( \frac{1}{2}(4t)^2 \sum_{j \in \Lambda_\delta^c} \Big( \sum_{i \in \Lambda_\delta} A_{ij} X_i \Big)^2 \right) = \mathbb{E}_g \exp\left( 4t \sum_{j \in \Lambda_\delta^c} g_j \Big( \sum_{i \in \Lambda_\delta} A_{ij} X_i \Big)^2 \right).$$

Rearranging the terms, and using that the resulting term is a linear combination of the sub-Gaussian $X_i$, $i \in \Lambda_\delta$, we find that

$$\mathbb{E}_{X,g} \exp\left( 4t \sum_{j \in \Lambda_\delta^c} g_j \Big( \sum_{i \in \Lambda_\delta} A_{ij} X_i \Big)^2 \right) = \mathbb{E}_{X,g} \exp\left( 4t \sum_{i \in \Lambda_\delta} X_i \Big( \sum_{j \in \Lambda_\delta^c} A_{ij} g_j \Big) \right)$$

$$\leq \mathbb{E}_{X,g} \exp\left( \frac{1}{2}(4t)^2 \sum_{i \in \Lambda_\delta} \Big( \sum_{j \in \Lambda_\delta^c} A_{ij} g_j \Big)^2 \right)$$

$$\leq \mathbb{E}_{X,g} \exp\left( 8t^2 \sum_i \Big( \sum_j \delta_i(1 - \delta_j) A_{ij} g_j \Big)^2 \right)$$

$$\leq \mathbb{E}_{X,g} \exp\left( 8t^2 \|A^\delta g\|_2^2 \right).$$

Now, by the rotation invariance of the distribution of $g$, the random variable $\|A^\delta g\|_2^2$ is distributed identically with $\sum_i s_i^2 g_i^2$ where $s_i$ denote the singular values of $A^\delta$, so by independence

$$\mathbb{E}_{X,g} \exp\left( 8t^2 \|A^\delta g\|_2^2 \right) = \mathbb{E}_g \exp\left( 8t^2 \sum_i s_i^2 g_i^2 \right)$$

$$= \prod_i \mathbb{E}_g \exp\left( 8t^2 s_i^2 g_i^2 \right)$$

$$\leq \prod_i \exp\left( 16t^2 s_i^2 \right)$$

where the last inequality holds if $8t^2 \max_i s_i^2 \leq \frac{1}{4}$ and arises since the MGF of a Chi-squared variable, $\mathbb{E}e^{tg^2} \leq e^{2t}$ for $0 \leq t \leq \frac{1}{4}$.

Since $\max_i s_i = \|A_\delta\| \leq \|A\|$ and $\sum_i s_i^2 = \|A_\delta\|_F^2 \leq \|A\|_F$, we combine the above steps to find that

$$\mathbb{E}_X \exp(tW) \leq \exp\left( 16t^2 \|A\|_F^2 \right) \quad \text{for } |t| \leq (4\sqrt{2}\|A\|)^{-1}.$$

$\square$

## C.2 Permutation Bounds for Two-Sample U-Statistics

The main technical result we use for the null bounds is the following. Here we assume the permutation-invariance property of the U statistic kernel that $h(x_1, x_2; y_1, y_2) = h(x_2, x_1; y_1, y_2) = h(x_1, x_2; y_2, y_1)$. This can always be ensured by symmetrizing the kernel, and is satisfied by the kernel used by $\widehat{\mathrm{MMD}}^2$ which we consider in this paper.

**Theorem 7.** *Fix combined sample $\boldsymbol{Z}$ of size $n + m$ with $n \leq m$ and let $h(x_1, x_2; y_1, y_2)$ be a two-sample U-statistic kernel as described above. Define*

$$U_k(\boldsymbol{Z}) := \frac{1}{n(n-1)m(m-1)} \sum_{(i,i') \in [n]_2} \sum_{(j,j') \in [m]_2} h(Z_i, Z_{i'}; Z_{n+j}, Z_{n+j'}).$$

*and*

$$\bar{U}_k(\boldsymbol{Z}) = \frac{1}{n(n-1)m(m-1)} \sup_{\sigma \in \mathfrak{S}_{n+m}} \sqrt{\sum_{(i,i') \in [n]_2} \left( \sum_{(j,j') \in [m]_2} h(Z_{\sigma(i)}, Z_{\sigma(i')}; Z_{\sigma(n+j)}, Z_{\sigma(n+j')}) \right)^2}.$$

*If $\sigma \in \mathrm{Uniform}(\mathfrak{S}_{n+m})$ is a random permutation of the dataset, then then for all $|t| < (4\sqrt{2}\bar{U}_k(\boldsymbol{Z}))^{-1}$,*

$$\mathbb{E}_\sigma \exp(tU_k(\sigma\boldsymbol{Z})) \leq \exp\left(t^2 \bar{U}_k^2(\boldsymbol{Z})\right).$$

**Theorem 8** (Normaliser Bound). *If $h(x, x'; y, y') = k(x, x') + k(y, y') - k(x, y') - k(x', y)$ then*

$$\bar{U}_k(\boldsymbol{Z})^2 \leq \frac{16}{n(n-1)} \widehat{N}_k(\boldsymbol{Z})$$

*where $n \leq m$ and*

$$\widehat{N}_k(\boldsymbol{Z}) = \frac{1}{n(n-1)} \sum_{(i,j) \in [n+m]_2} k(Z_i, Z_j)^2.$$

*Additionally, if $0 \leq k(x, x') \leq \kappa$ for all $x, x'$, then*

$$\bar{U}_k^2(\boldsymbol{Z}) \leq \frac{4\kappa^2}{n(n-1)}.$$

**Proof of Theorem 7.** Firstly, we make the following definitions based on the work of Kim et al. (2022): let $L = \{l_1, \ldots, l_n\}$ be an $n$-tuple drawn uniformly without replacement from $[m]$. Then for *any* fixed $\boldsymbol{Z}$

$$\mathbb{E}_L[\tilde{U}_k^L(\boldsymbol{Z})] = U_k(\boldsymbol{Z}) \tag{7}$$

where we define

$$\tilde{U}_k^L(\boldsymbol{Z}) := \frac{1}{n(n-1)} \sum_{(i,j) \in [n]_2} h(Z_i, Z_j; Z_{n+l_i}, Z_{n+l_j}).$$

Note that in the above we have used the invariance $h(x_1, x_2; y_1, y_2) = h(x_1, x_2; y_2, y_1)$.

Now additionally define $\zeta_i$ as i.i.d. Rademacher variables, and let $\sigma \sim \mathrm{Uniform}(\mathfrak{S}_{n+m})$. For any fixed $\boldsymbol{Z}$

$$\tilde{U}_k^L(\sigma\boldsymbol{Z}) =^d \tilde{U}_k^{L,\zeta}(\sigma\boldsymbol{Z}) \tag{8}$$

where we have defined

$$\tilde{U}_k^{L,\zeta}(\boldsymbol{Z}) := \frac{1}{n(n-1)} \sum_{(i,j) \in [n]_2} \zeta_i \zeta_j h(Z_i, Z_j; Z_{n+l_i}, Z_{n+l_j}).$$

This works because we can first define $\tilde{Z}_i = Z_i$ or $\tilde{Z}_i = Z_{n+l_i}$, each with probability $\frac{1}{2}$. The distribution of $\tilde{U}_k^L(\sigma\tilde{\boldsymbol{Z}}) =^d \tilde{U}_k^L(\sigma\boldsymbol{Z})$, and then using the symmetry of $k(x, y)$ in its arguments gives the equivalence Equation (8) (*c.f.* eq. 28, Kim et al., 2022).

Now for fixed $\boldsymbol{Z}$, combining Equations (7) and (8) and Jensen's inequality we gives

$$\mathbb{E}_\sigma \exp(tU_k(\sigma\boldsymbol{Z})) = \mathbb{E}_\sigma \exp(t\mathbb{E}_L[\tilde{U}_k^L(\sigma\boldsymbol{Z})|\sigma])$$

$$= \mathbb{E}_\sigma \exp(t\mathbb{E}_{L,\zeta}[\tilde{U}_k^{L,\zeta}(\sigma \boldsymbol{Z})|\sigma])$$

$$\leq \mathbb{E}_{\sigma,\zeta} \exp\left( t\mathbb{E}_L \left[ \frac{1}{n(n-1)} \sum_{(i,j)\in[n]_2} \zeta_i\zeta_j h(Z_{\sigma(i)}, Z_{\sigma(j)}; Z_{\sigma(n+l_i)}, Z_{\sigma(n+l_j)}) \middle| \sigma \right] \right)$$

$$= \mathbb{E}_{\sigma,\zeta} \exp\left( t\frac{1}{n(n-1)} \sum_{(i,j)\in[n]_2} \zeta_i\zeta_j \mathbb{E}_L \left[ h(Z_{\sigma(i)}, Z_{\sigma(j)}; Z_{\sigma(n+l_i)}, Z_{\sigma(n+l_j)}) \middle| \sigma \right] \right)$$

$$=: \mathbb{E}_{\sigma,\zeta} \exp\left( t\sum_{i=1}^n \sum_{j=1}^n \zeta_i\zeta_j A_{ij}^\sigma \right).$$

In the final step we have defined the matrix $A^\sigma$, with entries $A_{ij}^\sigma = (n(n-1))^{-1}\mathbb{E}_L\left[ h(Z_{\sigma(i)}, Z_{\sigma(j)}; Z_{\sigma(n+l_i)}, Z_{\sigma(n+l_j)})\middle|\sigma \right]$ for $i \neq j$ and $A_{ii} = 0$ for all $i$.

We note that $\zeta$ is independent of $\sigma$ and satisfies the conditions of Theorem 6, so applying this theorem we obtain that

$$\mathbb{E}_{\sigma,\zeta} \exp\left( t\sum_{i=1}^n \sum_{j=1}^n \zeta_i\zeta_j A_{ij}^\sigma \right)$$

$$\leq \mathbb{E}_\sigma \exp\left( 16t^2\|A^\sigma\|_F^2 \right) \qquad \text{for } |t| < (4\sqrt{2}\|A^\sigma\|)^{-1}$$

$$\leq \mathbb{E}_\sigma \exp\left( 16t^2\|A^\sigma\|_F^2 \right) \qquad \text{for } |t| < (4\sqrt{2}\|A^\sigma\|_F)^{-1}$$

$$\leq \exp\left( 16t^2 \sup_{\sigma\in\mathfrak{S}_{n+m}} \|A^\sigma\|_F^2 \right) \qquad \text{for } |t| < (4\sqrt{2} \sup_{\sigma\in\mathfrak{S}_{n+m}} \|A^\sigma\|_F)^{-1}$$

where in the second line we have used that $\|M\| \leq \|M\|_F$ for any matrix $M$.

Now note that

$$\|A^\sigma\|_F^2 = \sum_{i=1}^n \sum_{i'=1}^n (A_{ii'}^\sigma)^2$$

$$= \frac{1}{n^2(n-1)^2} \sum_{(i,i')\in[n]_2} (\mathbb{E}_L\left[ h(Z_{\sigma(i)}, Z_{\sigma(i')}; Z_{\sigma(n+l_i)}, Z_{\sigma(n+l_{i'})})\middle|\sigma \right])^2$$

$$= \frac{1}{n^2(n-1)^2} \sum_{(i,i')\in[n]_2} \left( \frac{1}{m(m-1)} \sum_{(j,j')\in[m]_2} h(Z_{\sigma(i)}, Z_{\sigma(i')}; Z_{\sigma(n+j)}, Z_{\sigma(n+j')}) \right)^2.$$

We complete the proof by noting that $\bar{U}_k(\boldsymbol{Z})^2 = \sup_{\sigma\in\mathfrak{S}_{n+m}} \|A^\sigma\|_F^2$. $\qquad\square$

**Proof of Theorem 8.** We have

$$\bar{U}_k^2(\boldsymbol{Z}) = \frac{1}{n^2(n-1)^2} \sum_{(i,i')\in[n]_2} \left( \frac{1}{m(m-1)} \sum_{(j,j')\in[m]_2} h(Z_{\sigma(i)}, Z_{\sigma(i')}; Z_{\sigma(n+j)}, Z_{\sigma(n+j')}) \right)^2$$

$$\leq \frac{1}{n^2(n-1)^2 m(m-1)} \sum_{(i,i')\in[n]_2} \sum_{(j,j')\in[m]_2} h(Z_{\sigma(i)}, Z_{\sigma(i')}; Z_{\sigma(n+j)}, Z_{\sigma(n+j')})^2$$

$$\leq \frac{4}{n^2(n-1)^2 m(m-1)} \sum_{(i,i')\in[n]_2} \sum_{(j,j')\in[m]_2} \left( k(Z_{\sigma(i)}, Z_{\sigma(i')})^2 + k(Z_{\sigma(n+j)}, Z_{\sigma(n+j')})^2 \right.$$

$$\left. + k(Z_{\sigma(i)}, Z_{\sigma(n+j')})^2 + k(Z_{\sigma(i')}, Z_{\sigma(n+j')})^2 \right)$$

$$= \frac{4}{n^2(n-1)^2 m(m-1)} \left( m(m-1) \sum_{(i,i')\in[n]_2} k(Z_{\sigma(i)}, Z_{\sigma(i')})^2 \right.$$

$$+ n(n-1) \sum_{(j,j') \in [m]_2} k(Z_{\sigma(n+j)}, Z_{\sigma(n+j')})^2$$

$$+ (m-1)(n-1) \sum_{1 \leq i \leq n} \sum_{1 \leq j \leq m} k(Z_{\sigma(i)}, Z_{\sigma(n+j')})^2$$

$$+ (m-1)(n-1) \sum_{1 \leq i \leq n} \sum_{1 \leq j \leq m} k(Z_{\sigma(i')}, Z_{\sigma(n+j)})^2 \Bigg)$$

$$\leq \frac{16}{n^2(n-1)^2} \sum_{(i,j) \in [n+m]_2} k(Z_i, Z_j)^2$$

$$= \frac{16}{n(n-1)} \widehat{N}_k(\boldsymbol{Z})$$

where the first inequality holds by Jensen's inequality, the second by convexity (so that $(a + b + c + d)^2 \leq 4(a^2 + b^2 + c^2 + d^2)$) and the third inequality using $n \leq m$ to upper bound each of the four scalars inside the parentheses by $m(m-1)$, and we also upper bounded each of sums over subsets of terms by the sum over all possible terms.

For the second part note that since $h \in [-2\kappa, 2\kappa]$

$$\bar{U}_k^2(\boldsymbol{Z}) = \frac{1}{n^2(n-1)^2} \sum_{(i,i') \in [n]_2} \left( \frac{1}{m(m-1)} \sum_{(j,j') \in [m]_2} h(Z_{\sigma(i)}, Z_{\sigma(i')}; Z_{\sigma(n+j)}, Z_{\sigma(n+j')}) \right)^2$$

$$\leq \frac{1}{n^2(n-1)^2} \sum_{(i,i') \in [n]_2} (2\kappa)^2$$

$$\leq \frac{4\kappa^2}{n(n-1)}.$$

$\square$

## C.3 Bounds for MMD-FUSE under Null: Proof of Theorem 5

**Proof of Theorem 5.** Note under the null $\boldsymbol{Z}$ is permutation invariant. We will use Theorem 7 with $h$ as the MMD 2-statistic kernel, so that $U_k(\boldsymbol{Z}) = \widehat{\text{MMD}}^2(\boldsymbol{Z})$ (for fixed $k$). We will use the notation $\leq_{1-\delta}$ to denote that the inequality holds with probability at least $1 - \delta$ over the random variables being considered.

Note that $\bar{U}_k^2 \leq 4\kappa^2/n(n-1)$ from Theorem 8, so that

$$\log \mathbb{E}_{k \sim \pi(\langle \boldsymbol{Z} \rangle)} \exp\left( \lambda \widehat{\text{MMD}}_k^2(\boldsymbol{X}, \boldsymbol{Y}) \right)$$

$$\leq_{1-\delta} \log \mathbb{E}_{\boldsymbol{Z}} \mathbb{E}_{k \sim \pi(\langle \boldsymbol{Z} \rangle)} \exp\left( \lambda \widehat{\text{MMD}}_k^2(\boldsymbol{X}, \boldsymbol{Y}) \right) + \log \frac{1}{\delta} \qquad \text{Markov}$$

$$= \log \mathbb{E}_{\sigma} \mathbb{E}_{\boldsymbol{Z}} \mathbb{E}_{k \sim \pi(\langle \boldsymbol{Z} \rangle)} \exp\left( \lambda \widehat{\text{MMD}}_k^2(\sigma \boldsymbol{Z}) \right) + \log \frac{1}{\delta} \qquad \text{Null Permutation-Free}$$

$$= \log \mathbb{E}_{\boldsymbol{Z}} \mathbb{E}_{k \sim \pi(\langle \boldsymbol{Z} \rangle)} \mathbb{E}_{\sigma} \exp\left( \lambda \widehat{\text{MMD}}_k^2(\sigma \boldsymbol{Z}) \right) + \log \frac{1}{\delta} \qquad \text{Prior Permutation-Free}$$

$$\leq \log \mathbb{E}_{\boldsymbol{Z}} \mathbb{E}_{k \sim \pi(\langle \boldsymbol{Z} \rangle)} \mathbb{E}_{\sigma} \exp\left( \lambda^2 \bar{U}_k^2 \right) + \log \frac{1}{\delta}$$

$$\leq \frac{4\kappa^2 \lambda^2}{n(n-1)} + \log \frac{1}{\delta}, \qquad (9)$$

where the penultimate result holds using Theorem 7 provided $|\lambda| < (4\sqrt{2} \sup_k \bar{U}_k(\boldsymbol{Z}))^{-1}$. Using the bound on $\bar{U}_k$ from above gives the $|\lambda| < \sqrt{n(n-1)}/8\sqrt{2}\kappa$ in the theorem statement.

By dividing $\widehat{\text{MMD}}^2$ by the permutation invariant $\widehat{N}_k$ in every one of the above, we also find that

$$\log \mathbb{E}_{k \sim \pi(\langle \boldsymbol{Z} \rangle)} \exp\left( \lambda \frac{\widehat{\text{MMD}}_k^2(\boldsymbol{X}, \boldsymbol{Y})}{\sqrt{\widehat{N}_k(\langle \boldsymbol{Z} \rangle)}} \right) \leq_{1-\delta} \log \mathbb{E}_{\boldsymbol{Z}} \mathbb{E}_{k \sim \pi(\langle \boldsymbol{Z} \rangle)} \mathbb{E}_{\sigma} \exp\left( \frac{\lambda^2 \bar{U}_k^2}{\widehat{N}_k(\langle \boldsymbol{Z} \rangle)} \right) + \log \frac{1}{\delta}$$

$$\leq \frac{16\lambda^2}{n(n-1)} + \log\frac{1}{\delta} \tag{10}$$

$$\tag{11}$$

the first inequality using that $\widehat{N}_k$ is permutation-invariant and holding for $|\lambda| < \sqrt{\sup_k \widehat{N}_k}(4\sqrt{2}\sup_k \bar{U}_k(\boldsymbol{Z}))^{-1}$. The final step uses that $\bar{U}_k^2 \leq 16\widehat{N}_k/n(n-1)$ from Theorem 8 which also shows that $|\lambda| < \sqrt{n(n-1)}/16\sqrt{2}$ is sufficient.

The upper tail bounds for $\widehat{\mathrm{FUSE}}_N$ and $\widehat{\mathrm{FUSE}}_1$ immediately follow from the above by dividing through $\lambda > 0$ and from their definitions.

**Lower Bounds.** For the lower tails,

$$-\widehat{\mathrm{FUSE}}_1(\boldsymbol{Z}) = \frac{1}{\lambda}\log\mathbb{E}_{k\sim\pi(\langle\boldsymbol{Z}\rangle)}\exp(\lambda\widehat{\mathrm{MMD}}_k^2(\boldsymbol{Z}))$$

$$\leq -\mathbb{E}_{k\sim\pi(\langle\boldsymbol{Z}\rangle)}[\widehat{\mathrm{MMD}}_k^2(\boldsymbol{Z})] \qquad\qquad \text{Jensen}$$

$$= \frac{1}{s}\log\circ\exp(\mathbb{E}_{k\sim\pi(\langle\boldsymbol{Z}\rangle)}[-s\widehat{\mathrm{MMD}}_k^2(\boldsymbol{Z})]) \qquad \text{introduce dummy } s>0$$

$$\leq \frac{1}{s}\log\circ\exp(\mathbb{E}_{k\sim\pi(\langle\boldsymbol{Z}\rangle)} - s\widehat{\mathrm{MMD}}_k^2(\boldsymbol{Z})) \qquad \text{Jensen on exp}$$

$$\leq_{1-\delta} \frac{4\kappa^2 s}{n(n-1)} + \frac{1}{s}\log\frac{1}{\delta},$$

where the last line follows from Equation (9) replacing $\lambda$ by dummy variable $-s$ for $0 < s < \sqrt{n(n-1)}/8\sqrt{2}\kappa$; we importantly note that this bound held for negative $\lambda$ by Theorem 7.

By following the same process for $\widehat{\mathrm{FUSE}}_N$ and instead using Equation (10), also holding for potentially negative $\lambda$, we obtain

$$-\widehat{\mathrm{FUSE}}_N(\boldsymbol{Z}) \leq_{1-\delta} \frac{16s}{n(n-1)} + \frac{1}{s}\log\frac{1}{\delta}$$

for $0 < s < \sqrt{n(n-1)}/16\sqrt{2}$.

For the small $\delta$ we generally use, these are tightest when $s$ is at its maximum permissible value, so we substitute these values for the theorem statement.

**Under Permutation.** To prove the equivalent result for $\sigma\boldsymbol{Z}$ under permutations, we replace $\boldsymbol{Z}$ by $\sigma\boldsymbol{Z}$ and our application of Markov's inequality introducing an expectation over $\boldsymbol{Z}$ with one over $\sigma$. This changes nothing else in the derivations. $\qquad\square$

## D   Concentration under the alternative

We give the exponential convergence bounds for $\widehat{\mathrm{FUSE}}_1$ under the alternative and relate it to its mean. In the following we will assume that $\mathcal{K}$ is a class of kernels bounded by $0 < \kappa < \infty$, so that $\widehat{\mathrm{FUSE}}_1 \in [-2\kappa, 2\kappa]$. We also introduce the following quantity (for fixed, data-free prior $\pi$) which is closely related to the expectation of $\widehat{\mathrm{FUSE}}_1$,

$$
\mathrm{FUSE}_1 := \sup_{\rho \in \mathcal{M}_+^1(\mathcal{K})} \mathrm{MMD}^2_{K_\rho}(p, q) - \frac{\mathrm{KL}(\rho, \pi)}{\lambda}.
$$

**Theorem 9.** $\mathrm{FUSE}_1$ *is bounded in the following ways:*

$$
\mathrm{FUSE}_1 \leq \mathbb{E}_{\boldsymbol{Z}} \widehat{\mathrm{FUSE}}_1(\boldsymbol{Z}) \leq \mathrm{FUSE}_1 + 8\kappa^2 \lambda \left( \frac{1}{n} + \frac{1}{m} \right),
$$

*and*

$$
\mathrm{MMD}^2_{K_\pi}(p, q) \leq \mathrm{FUSE}_1 \leq \sup_{\rho \in \mathcal{M}_+^1(\mathcal{K}) : \mathrm{KL}(\rho, \pi) < \infty} \mathrm{MMD}^2_{K_\rho}(p, q) \leq \sup_{k \in \mathrm{supp}(\pi)} \mathrm{MMD}^2_k(p, q).
$$

*Under the null hypothesis* $\mathrm{FUSE}_1 = 0$.

We can now state concentration results for $\widehat{\mathrm{FUSE}}_1$ in terms of $\mathrm{FUSE}_1$.

**Theorem 10.** *With probability at least* $1 - \delta$ *over the sample*

$$
\widehat{\mathrm{FUSE}}_1(\boldsymbol{Z}) - \mathrm{FUSE}_1 \leq 8\kappa^2 \lambda \left( \frac{1}{n} + \frac{1}{m} \right) + \frac{\log \delta^{-1}}{\lambda}
$$

*and with the same probability,*

$$
\mathrm{FUSE}_1 - \widehat{\mathrm{FUSE}}_1(\boldsymbol{Z}) \leq 2\kappa \sqrt{8 \left( \frac{1}{n} + \frac{1}{m} \right) \log \delta^{-1}}.
$$

### D.1   Proofs

**Note on the proofs.**   The "bounded difference lemma" (Theorem 11) we give below is not the same as the bounded difference inequality, though it is closely related. In fact, our result could be used as an intermediate step in proving the latter, but we note that this would not be the usual method (*e.g.* Boucheron et al., 2013, use an "entropy" method instead), and the converse is not true. The fact that we have to prove a variant form of an existing concentration inequality to get concentration for our log-sum-exp statistics is similar to how the same is required in PAC-Bayesian proofs, where *e.g.* PAC-Bayes Bernstein inequalities also require modified proof techniques that mirror those used to prove the usual Bernstein inequalities.

The usual bounded difference inequality cannot be used to prove Theorem 9, since it is not a concentration bound. It also does not give the concentration we need in Theorem 10, since the FUSE statistics do not have the bounded difference property, only $\widehat{\mathrm{MMD}}^2_k$ does; this is why our proofs cannot use a "plug-in" concentration bound.

**Theorem 11** (Bounded Difference Lemma). *A function $f$ has the* bounded difference *property there exist constants $L_\ell < \infty, \ell \in [n]$ such that*

$$
|f(z_1, \ldots, z_k, \ldots, z_n) - f(z_1, \ldots, z_\ell, \ldots, z_n)| \leq L_\ell
$$

*for any choices of $z_1, \ldots, z_n, z'_\ell$, and $\ell \in [n]$.*

*For such a function and any independent random variables $Z_1, \ldots, Z_n$ and $t \in \mathbb{R}$ (subject to appropriate measurability restrictions)*

$$
\mathbb{E} \exp(t(f(Z_1, \ldots, Z_n) - \mathbb{E} f(Z_1, \ldots, Z_n))) \leq \exp \left( \frac{1}{8} t^2 \sum_{\ell=1}^n L_\ell^2 \right).
$$

**Proof of Theorem 11.** We introduce the Doob construction, defining

$$D_\ell = \mathbb{E}[f(Z_1, \ldots, Z_n)|Z_1, \ldots, Z_\ell] - \mathbb{E}[f(Z_1, \ldots, Z_n)|Z_1, \ldots, Z_{\ell-1}]$$

This is a martingale difference sequence with

$$\sum_{\ell=1}^{n} D_\ell = f(Z_1, \ldots, Z_n) - \mathbb{E}f(Z_1, \ldots, Z_n).$$

It is shown by (for example) Wainwright (2019, Ch. 2.2, p.37) that $D_\ell$ lies in an interval of length at most $L_\ell$ by the bounded differences assumption. Thus, applying iterated expectation and Hoeffding's lemma for the MGF of bounded random variables,

$$\begin{aligned}
\mathbb{E}\exp(t(f(Z_1, \ldots, Z_n) - \mathbb{E}f(Z_1, \ldots, Z_n))) &= \mathbb{E}\exp\left(\lambda \sum_{\ell=1}^{n} D_\ell\right) \\
&= \mathbb{E}\left[\exp\left(\lambda \sum_{\ell=1}^{n-1} D_\ell\right) \mathbb{E}\left[\exp\left(\lambda D_n\right)|Z_1, \ldots, Z_{n-1}\right]\right] \\
&\leq \mathbb{E}\left[\exp\left(\lambda \sum_{\ell=1}^{n-1} D_\ell\right)\right] \exp\left(\frac{1}{8}\lambda^2 L_n^2 D_n\right) \\
&\leq \exp\left(\frac{1}{8}t^2 \sum_{\ell=1}^{n} L_\ell^2\right).
\end{aligned}$$

$\square$

The MGF of $\widehat{\mathrm{MMD}}^2$ can then be bounded using the following result, proved via the above.

**Theorem 12.** *For bounded kernel $k \leq \kappa$, $t \in \mathbb{R}$ and sample sizes $m, n$,*

$$\mathbb{E}\exp(t(\widehat{\mathrm{MMD}}_k^2(\boldsymbol{X}, \boldsymbol{Y}) - \mathrm{MMD}_k^2(p, q))) \leq \exp\left(8t^2\kappa^2\left(\frac{1}{m} + \frac{1}{n}\right)\right).$$

**Proof of Theorem 12.** We show that $\widehat{\mathrm{MMD}}_k^2(x, y)$ has the bounded differences property and then apply Theorem 11. Denote by $x^{\backslash \ell}$ for $\ell \in [n]$ that the $\ell$-th example in the $x$ sample is changed. Then

$$\begin{aligned}
&|\widehat{\mathrm{MMD}}_k^2(x, y) - \widehat{\mathrm{MMD}}_k^2(x^{\backslash \ell}, y)| \\
&= \left| \frac{2}{n(n-1)} \sum_{i \in [n]\backslash\{\ell\}} (k(x_\ell, x_i) - k(x_\ell', x_i)) - \frac{2}{mn} \sum_{j=1}^{m} (k(x_\ell, y_j) - k(x_\ell', y_j)) \right| \\
&\leq \frac{2}{n(n-1)} \sum_{i \in [n]\backslash\{\ell\}} |k(x_\ell, x_i) - k(x_\ell', x_i)| + \frac{2}{mn} \sum_{j=1}^{m} |k(x_\ell, y_j) - k(x_\ell', y_j)| \\
&\leq \frac{2}{n(n-1)}(n-1) \cdot 2\kappa + \frac{2}{mn} m \cdot 2\kappa \\
&= \frac{8\kappa}{n}.
\end{aligned}$$

A similar process for the $y$ sample gives bounds of $8\kappa/m$, so that $\widehat{\mathrm{MMD}}^2$ has the bounded differences property with

$$\sum_{\ell=1}^{n+m} L_\ell^2 \leq n \cdot \left(\frac{8\kappa}{n}\right)^2 + m \cdot \left(\frac{8\kappa}{m}\right)^2 = 64\kappa^2\left(\frac{1}{n} + \frac{1}{m}\right).$$

$\square$

**Proof of Theorem 9.** For the first lower bound, note

$$\mathbb{E}_{\boldsymbol{Z}}\widehat{\mathrm{FUSE}}_1(\boldsymbol{Z}) = \mathbb{E}_{\boldsymbol{Z}} \sup_\rho \mathbb{E}_\rho[\widehat{\mathrm{MMD}}^2] - \frac{\mathrm{KL}}{\lambda} \geq \sup_\rho \mathbb{E}_{\boldsymbol{Z},\rho}[\widehat{\mathrm{MMD}}^2] - \frac{\mathrm{KL}}{\lambda} = \mathrm{FUSE}_1.$$

For the upper bound, note that

$$\mathbb{E}_{\boldsymbol{Z}}\widehat{\mathrm{FUSE}}_1(\boldsymbol{Z}) \leq \left[\frac{1}{\lambda}\log\left(\mathbb{E}_{\boldsymbol{Z}}\mathbb{E}_{k\sim\pi}\left[e^{\lambda\widehat{\mathrm{MMD}}_k^2}\right]\right)\right] \qquad\text{Jensen}$$

$$\leq \left[\frac{1}{\lambda}\log\left(\mathbb{E}_{k\sim\pi}\mathbb{E}_{\boldsymbol{Z}}\left[e^{\lambda\widehat{\mathrm{MMD}}_k^2}\right]\right)\right] \qquad\text{Independence of }\pi\text{ from }\boldsymbol{Z}$$

$$\leq \frac{1}{\lambda}\log\left(\mathbb{E}_{k\sim\pi}\left[e^{\lambda\,\mathrm{MMD}_k^2 + 8\lambda^2\kappa^2(n^{-1}+m^{-1})}\right]\right) \qquad\text{Theorem 12}$$

$$= \mathrm{FUSE}_1 + 8\kappa^2\lambda\left(\frac{1}{n}+\frac{1}{m}\right).$$

The lower bound on $\mathrm{FUSE}_1$ is obtained by relaxing the supremum with $\rho = \pi$. The upper bounds come since the supremum over $\rho \in \mathcal{M}_+^1(\mathcal{K}) : \mathrm{KL}(\rho,\pi) < \infty$ of $\mathrm{MMD}^2$ is clearly greater than the KL-regularised version. Under the null hypothesis, $\mathrm{MMD}_k(p,q) = 0$ for every kernel (regardless of them being characteristic or not), so $\mathrm{FUSE}_1 = 0$. $\qquad\square$

**Proof of Theorem 10.** For the upper bound,

$$\widehat{\mathrm{FUSE}}_1(\boldsymbol{Z}) = \left[\frac{1}{\lambda}\log\left(\mathbb{E}_{k\sim\pi}\left[e^{\lambda\widehat{\mathrm{MMD}}_k^2}\right]\right)\right]$$

$$\leq_{1-\delta} \left[\frac{1}{\lambda}\log\left(\mathbb{E}_{\boldsymbol{Z}}\mathbb{E}_{k\sim\pi}\left[e^{\lambda\widehat{\mathrm{MMD}}_k^2}\right]\right)\right] + \frac{\log\delta^{-1}}{\lambda} \qquad\text{Markov}$$

$$\leq \left[\frac{1}{\lambda}\log\left(\mathbb{E}_{k\sim\pi}\mathbb{E}_{\boldsymbol{Z}}\left[e^{\lambda\widehat{\mathrm{MMD}}_k^2}\right]\right)\right] + \frac{\log\delta^{-1}}{\lambda} \qquad\text{Independence of }\pi\text{ from }\boldsymbol{Z}$$

$$\leq \frac{1}{\lambda}\log\left(\mathbb{E}_{k\sim\pi}\left[e^{\lambda\,\mathrm{MMD}_k^2 + 8\lambda^2\kappa^2(m^{-1}+n^{-1})}\right]\right) + \frac{\log\delta^{-1}}{\lambda} \qquad\text{Theorem 12}$$

$$= \mathrm{FUSE}_1 + 8\kappa^2\lambda\left(\frac{1}{n}+\frac{1}{m}\right) + \frac{\log\delta^{-1}}{\lambda}.$$

For the lower bound, let $\rho^*$ be the value of $\rho$ achieving the supremum in the dual form of $\mathrm{FUSE}_1$ (which we note is independent of the sample), so that

$$\mathrm{FUSE}_1 - \widehat{\mathrm{FUSE}}_1(\boldsymbol{Z}) = \sup_\rho\left\{\mathbb{E}_{\rho^*}[\mathrm{MMD}_k^2] - \frac{\mathrm{KL}(\rho,\pi)}{\lambda}\right\} - \sup_\rho\left\{\mathbb{E}_\rho[\widehat{\mathrm{MMD}}_k^2] - \frac{\mathrm{KL}(\rho,\pi)}{\lambda}\right\}$$

$$= \mathbb{E}_{\rho^*}[\mathrm{MMD}_k^2] - \frac{\mathrm{KL}(\rho^*,\pi)}{\lambda} - \sup_\rho\left\{\mathbb{E}_\rho[\widehat{\mathrm{MMD}}_k^2] - \frac{\mathrm{KL}(\rho,\pi)}{\lambda}\right\}$$

$$\leq \mathbb{E}_{\rho^*}[\mathrm{MMD}_k^2] - \frac{\mathrm{KL}(\rho^*,\pi)}{\lambda} - \left(\mathbb{E}_{\rho^*}[\widehat{\mathrm{MMD}}_k^2] - \frac{\mathrm{KL}(\rho^*,\pi)}{\lambda}\right)$$

$$= \mathbb{E}_{\rho^*}[\mathrm{MMD}_k^2 - \widehat{\mathrm{MMD}}_k^2]$$

$$= \mathrm{MMD}_{K_{\rho^*}}^2(p,q) - \widehat{\mathrm{MMD}}_{K_{\rho^*}}^2(\boldsymbol{Z}).$$

Now the above is for fixed kernel $K_{\rho^*}$ independent of the data, so by Markov's inequality and Theorem 12, for any $s > 0$ we find

$$\mathrm{MMD}_{K_{\rho^*}}^2(p,q) - \widehat{\mathrm{MMD}}_{K_{\rho^*}}^2(\boldsymbol{Z})$$

$$\leq_{1-\delta} s^{-1}\log\mathbb{E}_{\boldsymbol{Z}}\exp\left(s\left(\mathrm{MMD}_{K_{\rho^*}}^2(p,q) - \widehat{\mathrm{MMD}}_{K_{\rho^*}}^2(\boldsymbol{Z})\right)\right) + s^{-1}\log\delta^{-1}$$

$$\leq 8s\kappa^2\left(\frac{1}{m}+\frac{1}{n}\right) + s^{-1}\log\delta^{-1}.$$

Optimising for $s$ yields $s = \kappa^{-1}\sqrt{\log(1/\delta)}/\sqrt{8(1/m+1/n)}$ which gives the desired result. $\qquad\square$

# E    Power Analysis

## E.1    General Recipe for Power Analysis of Permutation Tests

The general outline for power analysis consists of the following: First we start with the Type II error probability. Then we attempt to upper bound it by iteratively applying the following simple lemma to the different terms.

**Monotonicity in High Probability.**    Let $X, Y$ be r.v.s, such that $X \leq Y$ w.p. $\geq 1 - \delta$. Then $\mathbb{P}(X \geq c) \leq \mathbb{P}(Y \geq c) + \delta$. This result can be applied both when $X \leq Y$ a.s. giving $\delta = 0$, or if $Y = a$ is constant.

Iteratively applying this lemma gives $\mathbb{P}(\text{Type II}) \leq N\delta$, and then we set $N\delta = \beta$.

As a first step in the above process we will use the following useful result (adapted from Kim et al., 2022) to convert from the random permutations we use in practice to the full set of permutation. This result shows that when $B$ is taken sufficiently large (roughly $\Omega(\alpha^{-2} \log(\beta^{-1}))$), the only changes to the final power results will be in constants multiplying $\alpha$ and $\beta$.

**Theorem 13** (From Randomised to Deterministic Permutations).    *Suppose $G = (g_1, \ldots, g_{B+1})$ consists of $B$ uniformly drawn permutations from $\mathcal{G}$, plus the identity permutation as $g_{B+1}$. Then*

$$\mathbb{P}\left(\tau(\mathbf{Z}) \leq \underset{1-\alpha, G}{\text{quantile }} \tau(g\mathbf{Z})\right) \leq \mathbb{P}\left(\tau(\mathbf{Z}) \leq \underset{1-\alpha_B, \mathcal{G}}{\text{quantile }} \tau(g\mathbf{Z})\right) + \delta$$

*where $1 - \alpha_B = \frac{B+1}{B}(1 - \alpha) + \sqrt{\frac{\log(2/\delta)}{2B}}$.*

*We note that provided $B \geq 8\alpha^{-2} \log(2/\delta)$, then $1 - \alpha_B \leq 1 - \alpha/2$.*

**Proof.**    We note the Dvoretzky–Kiefer–Wolfowitz inequality Dvoretzky et al. (1956); Massart (1990) for empirical CDF $F_n$ of $n$ samples from original CDF $F$:

$$\mathbb{P}\left(\sup_x |F_n(x) - F(x)| \geq t\right) \leq 2e^{-2nt^2} \tag{12}$$

for every $t > 0$.

The permutation CDF for a group $\mathcal{G}$ take the form

$$F_{\mathcal{G}}(x) = \frac{1}{|\mathcal{G}|} \sum_{g \in \mathcal{G}} \mathbf{1}\{\tau(g\mathbf{Z}) \leq x\}$$

and given sample $\widetilde{G} = (g_1, \ldots, g_B)$ i.i.d. uniformly from $\mathcal{G}$ (this excludes the identity permutation added to $G$), the empirical CDF is

$$\widehat{F}_{\widetilde{G}}(x) = \frac{1}{B} \sum_{g \in \widetilde{G}} \mathbf{1}\{\tau(g\mathbf{Z}) \leq x\}$$

We can also write $\widehat{F}_G(x)$ including the identity permutation as $g_{B+1}$, and note that $\widehat{F}_G(x) =$

Define the good event

$$\mathcal{A} = \left\{\sup_x |F_{\widetilde{G}}(x) - F_{\mathcal{G}}(x)| \leq \sqrt{\frac{\log(2/\delta)}{2B}}\right\}$$

which holds with probability at least $1 - \delta$ by Equation (12). Given $\mathcal{A}$, we have

$$q_G := \underset{1-\alpha, g \in G}{\text{quantile }} \tau(g\mathbf{Z}) = \inf\left\{r \in \mathbb{R} : \frac{1}{B+1} \sum_{g \in G} \mathbf{1}\{\tau(g\mathbf{Z}) \leq r\} \geq 1 - \alpha\right\}$$

$$\leq \inf\left\{r \in \mathbb{R} : \frac{1}{B+1} \sum_{g \in G} \mathbf{1}\{\tau(g\mathbf{Z}) \leq r\} \geq 1 - \alpha\right\}$$

$$= \inf\{r \in \mathbb{R} : \widehat{F}_{\widetilde{G}}(r) \geq \frac{B+1}{B}(1-\alpha)\}$$

$$\leq \inf\left\{r \in \mathbb{R} : \widehat{F}_{\mathcal{G}}(r) \geq \frac{B+1}{B}(1-\alpha) + \sqrt{\frac{\log(2/\delta)}{2B}}\right\}$$

$$= \operatorname*{quantile}_{1-\alpha_B, g \in \mathcal{G}} \tau(g\boldsymbol{Z}) =: q$$

where we have defined $\alpha_B$ as above.

Overall we find $\mathbb{P}(\tau \leq q_G) \leq \mathbb{P}(\tau \leq q_G|\mathcal{A}) + \mathbb{P}(\mathcal{A}^c) \leq \mathbb{P}(\tau \leq q) + \delta$. $\qquad\square$

## E.2 Variance of $\widehat{\mathrm{MMD}}_k^2$

In proving our power results it is necessary to upper bound the variance of $\widehat{\mathrm{MMD}}_k^2$.

**Theorem 14.** *For any kernel $k$ upper bounded by $\kappa$, if $n \leq m \leq cn$ for $c \geq 1$, there exists universal constant $C > 0$ depending only on $c$, such that*

$$\mathbb{V}[\widehat{\mathrm{MMD}}_k^2] \leq C\left(\frac{4\kappa\,\mathrm{MMD}_k^2}{n} + \frac{\kappa^2}{n^2}\right)$$

**Proof.** We define

$$\sigma_{10}^2 = \mathbb{V}_X\left(\mathbb{E}_{X',Y,Y'}[h(X,X',Y,Y')]\right)$$
$$\sigma_{01}^2 = \mathbb{V}_Y\left(\mathbb{E}_{X,X',Y'}[h(X,X',Y,Y')]\right)$$
$$\sigma_{11}^2 = \max\{\mathbb{E}[k^2(X,X')], \mathbb{E}[k^2(X,Y)], \mathbb{E}[k^2(Y,Y')]\}.$$

From a well-known bound (based on Lee, 1990, Equation 2, p.38; see also Kim et al., 2022, Appendix F, Equation 59 or Schrab et al., 2023, Proposition 3),

$$\mathbb{V}[\widehat{\mathrm{MMD}}_k^2] \leq C\left(\frac{\sigma_{10}^2}{m} + \frac{\sigma_{01}^2}{n} + \sigma_{11}^2\left(\frac{1}{m} + \frac{1}{n}\right)^2\right)$$

$$\leq C\left(\frac{\sigma_{10}^2}{n} + \frac{\sigma_{01}^2}{n} + \frac{\sigma_{11}^2}{n^2}\right)$$

where we used that $n \leq m \leq cn$. and the result in red above, and the boundedness of the kernel for the final term. This gives the further bound

$$\mathbb{V}[\widehat{\mathrm{MMD}}_k^2] \leq C\left(\frac{4\kappa\,\mathrm{MMD}_k^2}{n} + \frac{\kappa^2}{n^2}\right)$$

since

$$\sigma_{10}^2 = \operatorname{var}_X\left(\mathbb{E}_{X',Y,Y'}[h(X,X',Y,Y')]\right)$$
$$= \mathbb{E}_X\left[\left(\mathbb{E}_{X',Y,Y'}[h(X,X',Y,Y')]\right)^2\right]$$
$$= \mathbb{E}_X\left[\left(\mathbb{E}_{X',Y,Y'}[\langle\phi(X)-\phi(Y),\phi(X')-\phi(Y')\rangle]\right)^2\right]$$
$$= \mathbb{E}_X\left[\langle\phi(X)-\mu_Q,\mu_P-\mu_Q\rangle^2\right]$$
$$\leq \left(\mathbb{E}_X\left[\|\phi(X)-\mu_Q\|^2\right]\right)\|\mu_P-\mu_Q\|^2$$
$$\leq 2\kappa\|\mu_P-\mu_Q\|^2$$
$$= 2\kappa\,\mathrm{MMD}_k^2,$$

a similar result for $\sigma_{01}$, and the simple bound $\sigma_{11}^2 \leq \kappa^2$. $\qquad\square$

### E.3 Proof of Theorem 2

**Proof.** Define $\alpha_B$ as in Theorem 13, noting that $1 - \alpha_B < 1 - \alpha/2$ under the assumption $B \geq 8\alpha^{-2} \log(4/\beta)$. From Theorem 13 we can consider the full permutation set as

$$\mathbb{P}_{p \times q, G} \left( \widehat{\mathrm{FUSE}}_1(\boldsymbol{Z}) \leq \underset{1-\alpha,G}{\mathrm{quantile}} \, \widehat{\mathrm{FUSE}}_1(\sigma \boldsymbol{Z}) \right)$$

$$\leq \mathbb{P}_{p \times q} \left( \widehat{\mathrm{FUSE}}_1(\boldsymbol{Z}) \leq \underset{1-\alpha_B,\mathcal{G}}{\mathrm{quantile}} \, \widehat{\mathrm{FUSE}}_1(\sigma \boldsymbol{Z}) \right) + \beta/2.$$

We also recall that from Theorem 5 when $\lambda = cn/\kappa$,

$$\underset{1-\alpha_B,\mathcal{G}}{\mathrm{quantile}} \, \widehat{\mathrm{FUSE}}_1(\sigma \boldsymbol{Z}) \leq \frac{C_1 \kappa (1 + \log \alpha^{-1})}{n},$$

so that

$$\mathbb{P}_{p \times q} \left( \widehat{\mathrm{FUSE}}_1(Z) \leq \underset{1-\alpha_B,\mathcal{G}}{\mathrm{quantile}} \, \widehat{\mathrm{FUSE}}_1(\sigma \boldsymbol{Z}) \right) \leq \mathbb{P}_{p \times q} \left( \widehat{\mathrm{FUSE}}_1(\boldsymbol{Z}) \leq \frac{C_1 \kappa (1 + \log \alpha^{-1})}{n} \right).$$

For any $\rho$, we define

$$S_\rho = \mathrm{MMD}^2_{K_\rho}(p, q) - \frac{\mathrm{KL}(\rho, \pi)}{\lambda} - \frac{C_1 \kappa (1 + \log \alpha^{-1})}{n},$$

which we substitute into

$$\mathbb{P}_{p \times q} \left( \widehat{\mathrm{FUSE}}_1(\boldsymbol{Z}) \leq \frac{C_1 \kappa (1 + \log \alpha^{-1})}{n} \right)$$

$$= \mathbb{P} \left( \mathrm{MMD}^2_{K_\rho}(p, q) - \frac{1}{\lambda} \mathrm{KL}(\rho, \pi) - \widehat{\mathrm{FUSE}}_1(\boldsymbol{Z}) \geq S_\rho \right)$$

$$= \mathbb{P} \left( \mathrm{MMD}^2_{K_\rho}(p, q) - \frac{1}{\lambda} \mathrm{KL}(\rho, \pi) - \sup_{\rho'} \left( \widehat{\mathrm{MMD}}^2_{K_{\rho'}}(\boldsymbol{Z}) - \frac{1}{\lambda} \mathrm{KL}(\rho', \pi) \right) \geq S_\rho \right)$$

$$\leq \mathbb{P} \left( \mathrm{MMD}^2_{K_\rho}(p, q) - \widehat{\mathrm{MMD}}^2_{K_\rho}(\boldsymbol{Z}) \geq S_\rho \right)$$

$$= \frac{1}{S_\rho^2} \mathbb{V}_{p \times q} \left[ \widehat{\mathrm{MMD}}^2_{K_\rho}(\boldsymbol{Z}) \right]$$

$$\leq \frac{C_2}{S_\rho^2} \left( \frac{4 \kappa \, \mathrm{MMD}^2}{n} + \frac{\kappa^2}{n^2} \right).$$

After substituting $S_\rho$, we used the dual form of $\widehat{\mathrm{FUSE}}_1$ and the inequalities $\sup f(\rho') \geq f(\rho)$, Chebyshev's, and Theorem 14. This term is upper bounded by $\beta/2$ if we set

$$S_\rho^2 > \frac{2C_2}{\beta} \left( \frac{4 \kappa \, \mathrm{MMD}^2}{n} + \frac{\kappa^2}{n^2} \right).$$

We also note that for $a, b, x$ all non-negative, if $x^2 > a^2 + 2b$, then $x^2 > ax + b$. This works because $x^2 > ax + b$ is equivalent to $x^2 > \left( \frac{a}{2} + \sqrt{\frac{a^2}{4} + b} \right)^2$ by taking the positive root, and

$$\left( \frac{a}{2} + \sqrt{\frac{a^2}{4} + b} \right)^2 = \frac{a^2}{2} + b + 2 \frac{a}{2} \sqrt{\frac{a^2}{4} + b} \leq a^2 + 2b$$

using Young's inequality $2AB \leq A^2 + B^2$.

Combining the above the Type II error rate is controlled by $\beta$ provided any of the following statements are true for any $\rho$ (with each new result implying the former):

$$\mathrm{MMD}^2_{K_\rho}(p, q) > \frac{\mathrm{KL}(\rho, \pi)}{\lambda} + \frac{C_1 \kappa (1 + \log \alpha^{-1})}{n} + \sqrt{\frac{2C_2}{\beta} \left( \frac{4 \kappa \, \mathrm{MMD}^2}{n} + \frac{\kappa^2}{n^2} \right)}$$

$$\mathrm{MMD}^2_{K_\rho}(p,q) > \frac{\mathrm{KL}(\rho,\pi)}{\lambda} + \frac{C_1\kappa(1+\log\alpha^{-1})}{n} + \sqrt{\frac{8C_2\kappa}{n\beta}}\,\mathrm{MMD} + \frac{\sqrt{2C_2}\kappa}{n\sqrt{\beta}}$$

$$\mathrm{MMD}^2_{K_\rho}(p,q) > \frac{2\,\mathrm{KL}(\rho,\pi)}{\lambda} + \frac{2C_1\kappa(1+\log\alpha^{-1})}{n} + \frac{8C_2\kappa}{n\beta} + \frac{2\sqrt{2C_2}\kappa}{n\sqrt{\beta}}$$

$$\mathrm{MMD}^2_{K_\rho}(p,q) > \frac{C_3\kappa}{n}\left(\frac{1}{\beta} + \log\frac{1}{\alpha} + \mathrm{KL}(\rho,\pi)\right)$$

where we used that $\sqrt{x+y} \le \sqrt{x} + \sqrt{y}$, the result above, and $\lambda = cn/\kappa$. $\qquad\square$

### E.4 Proof of Theorem 3

The proof of this statement proceeds similarly to the proof of Theorem 2. Using Theorem 5 with $\lambda = cn$ and Theorem 13 we find that

$$\mathbb{P}_{p\times q,G}\left(\widehat{\mathrm{FUSE}}_N(\boldsymbol{Z}) \le \operatorname*{quantile}_{1-\alpha,G}\widehat{\mathrm{FUSE}}_N(\sigma\boldsymbol{Z})\right)$$

$$\le \mathbb{P}_{p\times q}\left(\widehat{\mathrm{FUSE}}_N(\boldsymbol{Z}) \le \frac{C_1(1+\log\alpha^{-1})}{n}\right) + \beta/2.$$

For any $\rho$ such that $\mathrm{KL}(\rho,\pi) < \infty$ we define

$$T_\rho = \frac{C_1}{\kappa}\cdot\mathrm{MMD}^2_{K_\rho}$$

$$S_\rho = T_\rho - \frac{1}{\lambda}\mathrm{KL}(\rho,\pi) - \frac{C_1\kappa(1+\log\alpha^{-1})}{n}.$$

We assumed that $n \le m \le cn$ for some $c \ge 1$. Based on this, note that $N_k \le \kappa^2/C_1^2$ for $C_1$ depending only on $m/n \in [1,c]$, and (since $\mathrm{MMD}^2 \ge 0$ is strictly non-negative, unlike $\widehat{\mathrm{MMD}}^2$ which can be negative),

$$-\mathbb{E}_\rho\left[\frac{\mathrm{MMD}^2_k}{\sqrt{\widehat{N}_k(\boldsymbol{Z})}}\right] \le -\frac{C_1}{\kappa}\mathbb{E}_{k\sim\rho}\left[\mathrm{MMD}^2_k\right] = -T_\rho. \qquad (13)$$

Now we introduce these definitions to bound

$$\mathbb{P}\left(\widehat{\mathrm{FUSE}}_N(\boldsymbol{Z}) \le \frac{C_1(1+\log\alpha^{-1})}{n}\right)$$

$$= \mathbb{P}\left(T_\rho - \frac{1}{\lambda}\mathrm{KL}(\rho,\pi) - \widehat{\mathrm{FUSE}}_N(\boldsymbol{Z}) \ge S_\rho\right)$$

$$= \mathbb{P}\left(T_\rho - \frac{1}{\lambda}\mathrm{KL}(\rho,\pi) - \sup_{\rho'}\left(\mathbb{E}_{k\sim\rho'}\left[\frac{\widehat{\mathrm{MMD}}^2_k(\boldsymbol{Z})}{\sqrt{\widehat{N}_k(\boldsymbol{Z})}}\right] - \frac{1}{\lambda}\mathrm{KL}(\rho',\pi)\right) \ge S_\rho\right)$$

$$\le \mathbb{P}\left(T_\rho - \mathbb{E}_{k\sim\rho}\left[\frac{\widehat{\mathrm{MMD}}^2_k(\boldsymbol{Z})}{\sqrt{\widehat{N}_k(\boldsymbol{Z})}}\right] \ge S_\rho\right) \qquad\qquad \sup f(\rho') \ge f(\rho)$$

$$= \mathbb{P}\left(T_\rho - \mathbb{E}_{k\sim\rho}\left[\frac{\mathrm{MMD}^2_k}{\sqrt{\widehat{N}_k(\boldsymbol{Z})}} - \frac{\mathrm{MMD}^2_k}{\sqrt{\widehat{N}_k(\boldsymbol{Z})}} + \frac{\widehat{\mathrm{MMD}}^2_k(\boldsymbol{Z})}{\sqrt{\widehat{N}_k(\boldsymbol{Z})}}\right] \ge S_\rho\right)$$

$$= \mathbb{P}\left(T_\rho - \mathbb{E}_{k\sim\rho}\left[\frac{\mathrm{MMD}^2_k}{\sqrt{\widehat{N}_k(\boldsymbol{Z})}}\right] + \mathbb{E}_{k\sim\rho}\left[\frac{\mathrm{MMD}^2_k - \widehat{\mathrm{MMD}}^2_k(\boldsymbol{Z})}{\sqrt{\widehat{N}_k(\boldsymbol{Z})}}\right] \ge S_\rho\right)$$

$$= \mathbb{P}\left(T_\rho - \mathbb{E}_{k\sim\rho}\left[\frac{\mathrm{MMD}^2_k}{\sqrt{\widehat{N}_k(\boldsymbol{Z})}}\right] + \mathbb{E}_{k\sim\rho}\left[\frac{\mathrm{MMD}^2_k - \widehat{\mathrm{MMD}}^2_k(\boldsymbol{Z})}{\sqrt{\widehat{N}_k(\boldsymbol{Z})}}\right] \ge S_\rho\right)$$

$$\leq \mathbb{P}\left(\mathbb{E}_{k\sim\rho}\left[\frac{\mathrm{MMD}_k^2 - \widehat{\mathrm{MMD}}_k^2(\boldsymbol{Z})}{\sqrt{\widehat{N}_k(\boldsymbol{Z})}}\right] \geq S_\rho\right) \qquad\qquad \text{by Equation (13)}$$

$$\leq \mathbb{P}\left(\left|\mathbb{E}_{k\sim\rho}\left[\frac{\mathrm{MMD}_k^2 - \widehat{\mathrm{MMD}}_k^2(\boldsymbol{Z})}{\sqrt{\widehat{N}_k(\boldsymbol{Z})}}\right]\right| \geq S_\rho\right) \qquad\qquad x \leq |x|$$

$$\leq \mathbb{P}\left(\mathbb{E}_{k\sim\rho}\left[\left|\frac{\mathrm{MMD}_k^2 - \widehat{\mathrm{MMD}}_k^2(\boldsymbol{Z})}{\sqrt{\widehat{N}_k(\boldsymbol{Z})}}\right|\right] \geq S_\rho\right) \qquad\qquad |x| \text{ convex, Jensen}$$

$$= \mathbb{P}\left(\mathbb{E}_{k\sim\rho}\left[\frac{|\mathrm{MMD}_k^2 - \widehat{\mathrm{MMD}}_k^2(\boldsymbol{Z})|}{\sqrt{\widehat{N}_k(\boldsymbol{Z})}}\right] \geq S_\rho\right) \qquad\qquad \widehat{N}_k \text{ positive}$$

$$\leq \frac{1}{S_\rho}\mathbb{E}_{\boldsymbol{Z}}\mathbb{E}_{k\sim\rho}\left[\frac{|\mathrm{MMD}_k^2 - \widehat{\mathrm{MMD}}_k^2(\boldsymbol{Z})|}{\sqrt{\widehat{N}_k(\boldsymbol{Z})}}\right] \qquad\qquad \widehat{N}_k \text{ positive, Markov}$$

$$\leq \frac{1}{S_\rho}\sqrt{\mathbb{E}_{\boldsymbol{Z}}\mathbb{E}_{k\sim\rho}\left[|\mathrm{MMD}_k^2 - \widehat{\mathrm{MMD}}_k^2(\boldsymbol{Z})|^2\right]\mathbb{E}_{\boldsymbol{Z}}\mathbb{E}_{k\sim\rho}\left[\frac{1}{\widehat{N}_k(\boldsymbol{Z})}\right]} \qquad\qquad \text{Cauchy-Schwarz.}$$

Thus the type II error will be controlled by $\beta$ if for any $\rho$

$$S_\rho > \frac{2}{\beta}\sqrt{\mathbb{E}_{\boldsymbol{Z}}\mathbb{E}_{k\sim\rho}\left[|\mathrm{MMD}_k^2 - \widehat{\mathrm{MMD}}_k^2(\boldsymbol{Z})|^2\right]\mathbb{E}_{\boldsymbol{Z}}\mathbb{E}_{k\sim\rho}\left[\frac{1}{\widehat{N}_k(\boldsymbol{Z})}\right]}.$$

Provided there is a $c > 0$ with

$$\mathbb{E}_{p\times q}\left[\frac{1}{\widehat{N}_k(\boldsymbol{Z})}\right] \leq c$$

for all $k$, this reduces to the condition

$$\mathrm{MMD}_{K_\rho}^2(p, q) > C_2\kappa\left(\frac{\log \alpha^{-1}}{n} + \frac{1}{\beta}\sqrt{\mathbb{E}_\rho\mathbb{V}_{p\times q}\left[\widehat{\mathrm{MMD}}_k^2(\boldsymbol{Z})\right]} + \frac{\mathrm{KL}(\rho, \pi)}{n}\right).$$

Applying Theorem 14, the proof is completed in essentially the same way as in the result for $\widehat{\mathrm{FUSE}}_1$ (with slightly different $\beta$ dependence).

