# OpenReview forum: "MMD-Fuse: Learning and Combining Kernels for Two-Sample Testing Without Data Splitting"
_NeurIPS.cc/2023/Conference — NeurIPS 2023 spotlight_

### Official Review · Reviewer_fFeS · 2023-06-15

**Soundness:** 3 good
**Presentation:** 4 excellent
**Contribution:** 3 good
**Rating:** 7
**Confidence:** 4

**Summary:**

[Update: After intensive discussion with the authors, I changed my score from 6 to 7. I further increase the score for "Contribution" from 2 to 3.]

The paper introduces two new test statistics for permutation-based two-sample tests that are related to the MMD -- called FUSE_N and FUSE_1. These are motivated by selecting/combining good kernels for an MMD-based two sample test *without splitting* the data.
The authors theoretically show that FUSE_1 corresponds to a (regularized) supremum of the MMD over the possible kernel combinations. FUSE_N, which empirically performs much stronger, is derived from FUSE_1 but where the kernels are weighted by their normalized (squared) MMD estimates instead of simply by their (squared) MMD estimates. There is no theoretical justification for using FUSE_N.

The authors provide a theoretical power analysis of the proposed tests. Furthermore they run experiments that show their test has competitive power against a recent SOTA MMDAgg, while having a constant speed-up. The code is provided in a well-structured repository, such that it can be easily reproduced or the community can build on it.

The paper also gives some general insights into what test selection strategies are generally possible when using a permutation test. Whilst this might not be very new in itself, having a clear presentation of the general thoughts in one place, might be very useful for scientist entering the field.

**Strengths:**

- The delivered experiments are done very thoroughly and structured and the code is provided in a high quality.
- The paper is very well written and clarifies important (although not really new) aspects of permutation-based tests in an accessible way.
- The proposed test convinces in terms of power and runtime -- (although I am not 100% convinced by the speed-up over MMDAgg yet).
- Originality and significance: Two-sample testing is an ongoing research area and improvements and new ideas are published at NeurIPS and similar conferences regularly. The community seems nevertheless relatively small. The test statistic seems new, although its motivation is not really clear.

**Weaknesses:**

- The motivation for the proposed test statistic is not clear enough for me. FUSE_1 has somewhat a motivation (Sec. 4.1.) but:
  - it does not perform well and also the motivation does not arise from test power considerations.
  - It is well-known that selecting the kernel by maximizing the MMD is not a theoretically justified strategy (not scale invariant for example).
- Sutherland et al (2017) give a criterion for optimizing the kernel in terms of asymptotic power, which results in optimizing a Signal-to-noise ratio. The normalized statistic FUSE_N seems to go in this direction, but there is no motivation for the normalization given.
- Generally I am not really understanding the starting point for this paper. What motivated you to do this work and what where you trying to solve? In particular what is the similarity/difference to MMDAgg?
- The test statistic has a parameter $\lambda$. For the theoretical results, $\lambda$ depends on the sample size. However, in the experiments (as far as I understood) $\lambda=1$ irrespectively of the sample size. How was this value determined and where is it discussed? And what is the recommended way to do this in practice?

**Questions:**

Please also see my points in the weakness section.
### Questions I would like to discuss / understand:
- What is the motivation for FUSE_N (except that it works well on your experiments)?
- In Section A.4. you discuss the time-complexity relative to MMDAgg:
  - The difference is a factor of 2, so I do not think that the $\mathcal{O}$ notation is adequate here.
  - I also experienced that computing the kernel matrix is quite costly, compared to the permutations. But this only needs to be done once in both methods. Could it be that this difference is only visible because you used a very large number of permutations (20,000)?
- l. 143 "without incurring Bonferrony-type multiple testing penalties": I understand that your method does not have such a penalty, but what exactly is it that leads to it?
- What's the "N" or "1" in the naming of FUSE_ standing for?


### Questions / Remarks that do not require a response:
- l.57 "Our unsupervised feature extraction method is extemely powerful": I do not see any support for this statement. I agree that it is very general, but whether it works well, I do not see evidence in the paper.
- l.108f: Note that Kübler et al (AutoML Two-Sample Test) show in App. B.1 that the asymptotically optimal kernel for a fixed problem is indeed 1-dimensional.
- l. 116: I do not think that permuting the samples is exactly the same as sampling from 1/2 (p+q). In the permutation there are always exactly half the samples still from p and half from q. Whilst drawing from 1/2(p+q) can by chance (and most likely will) result unbalanced.

**Limitations:**

nothing specific

---

> ### Author Rebuttal · Authors · 2023-08-08
>
> **Reply to reviewer fFeS Part 1**
>
> Thank you for your valuable feedback and questions which will help us improve the presentation of our ideas.
>
>
> ## Permutation tests
> > The paper is very well written and clarifies important (although not really new) aspects of permutation-based tests in an accessible way.
>
> We are glad you found the paper to be well written, and that it clarifies important aspects of permutation-based tests. We would like to emphasise that although the result of Hemerik and Goeman is not a new one, the consequences of that result appear not to have been previously fully understood in the field. The simplicity and generality of our proposal to use permutation-invariant tuning of the statistic might lead one to expect that this idea has been used before, but to our best knowledge it has not. This is underlined by the fact that the median heuristic has been widely used without a formal justification, when one follows so straightforwardly from Theorem 1; or by the fact that the well-developed techniques of unsupervised and self-supervised learning have not been combined with two-sample permutation testing before, again despite this potential usage being a simple mathematical corollary of Theorem 1. We believe this latter fact alone has the potential to be highly impactful in this field.
>
>
> ## Motivation
> > The motivation for the proposed test statistic is not clear enough for me.
>
> > Sutherland et al (2017) give a criterion for optimising the kernel in terms of asymptotic power ... FUSE_N seems to go in this direction, but there is no motivation for the normalisation given.
>
> > Generally I am not really understanding the starting point for this paper. What motivated you to do this work and what where you trying to solve? In particular what is the similarity/difference to MMDAgg?
>
> Please see the overall discussion and motivation in the response to all reviewers.
>
>
>
>
> ## Time complexity
>
> > I also experienced that computing the kernel matrix is quite costly, compared to the permutations. But this only needs to be done once in both methods. Could it be that this difference is only visible because you used a very large number of permutations (20,000)?
>
> Using our code in Jax with a GPU, we find that computing the permuted MMD values is significantly more costly than computing the Gaussian kernel matrix (and this even for only 11 permutations). This explains why we observe that MMD-FUSE *always* runs twice as fast as MMDAgg.
>
> 1)
> - Sample size $m = n = 100$
> - Dimension $d = 100$
> - Compute kernel matrix: 3.8 ms
> - Given kernel matrix, compute MMD for $B+1$ permutations
> - - $B = 10$: 35 ms
> - - $B = 100$: 65 ms
> - - $B = 1000$: 573 ms
> - - $B = 2000$: 983 ms
>
> 2)
> - Sample size $m = n = 500$
> - Dimension $d = 100$
> - Compute kernel matrix: 3.82 ms
> - Given kernel matrix, compute MMD for $B+1$ permutations
> - - $B = 10$: 22 ms
> - - $B = 100$: 278 ms
> - - $B = 1000$: 2110 ms
> - - $B = 2000$: 4250 ms
>
> 3)
> - Sample size $m = n = 1000$
> - Dimension $d = 100$
> - Compute kernel matrix: 4.87 ms
> - Given kernel matrix, compute MMD for $B+1$ permutations
> - - $B = 10$: 71 ms
> - - $B = 100$: 555 ms
> - - $B = 1000$: 4240 ms
> - - $B = 2000$: 8490 ms
>
> 4)
> - Sample size $m = n = 2500$
> - Dimension $d = 100$
> - Compute kernel matrix: 29.6 ms
> - Given kernel matrix, compute MMD for $B+1$ permutations
> - - $B = 10$: 161 ms
> - - $B = 100$: 1170 ms
> - - $B = 1000$: 10800 ms
>
>
>
>
>
>
> > In Section A.4. you discuss the time-complexity relative to MMDAgg:
> > The difference is a factor of 2, so I do not think that the $O$ notation is adequate here.
>
> Agreed, this will be clarified in the final version.
>
>
>
>
>
>
> ## Bonferroni
>
> > l. 143 "without incurring Bonferroni-type multiple testing penalties"
>
> Such a penalty usually arises through multiple testing (where a correction is needed to ensure the correct threshold for all of the tests). More closely to our setup, it could also arise when looking at the concentration of a maximum of sub-Gaussian variables; specifically a term of $O(\sqrt{\log(|K|)})$ would be added when considering the maximum of multiple MMD values. This term would be carried over into the power expression, giving rise to the same penalty there as would arise in standard multiple testing. In the worst case for our method, where only a single kernel gives rise to a large normalised MMD and the rest give approximately zero normalised MMD, we would incur such a penalty; but in the case that several kernels do so the penalty would be considerably smaller. This makes testing with very large or infinite sets of kernels possible (though not always computationally practical).
>
>
>
> ## Miscellanea
>
> > $\lambda=1$ ?
>
> In the implementation, the parameter $\lambda$ is set to $\lambda = \sqrt{n(n-1)}$ (see line 222 of mmdfuse.py) which is chosen so that the two terms on the RHS of the FUSE bound in Theorem 5 are of the same order. We chose to define $\lambda$ only in terms of the sample size (ignoring constants). This value of $\lambda$ is of the order $n$ which is used to derive the power results l.929. We will include this explanation in Appendix A.2 "Test parameters".
>
> The confusion with $\lambda=1$ may have arisen from the fact that in our code we defined lambda = sqrt(m+n) * lambda_multiplier and have lambda_multiplier as a parameter with default value 1.
>
> > What's the "N" or "1" in the naming of FUSE_ standing for?
>
> The N for FUSE_N is not a parameter, it simply stands for 'Normalised'. The unnormalised version is referred to as FUSE_1 as it can be thought of as FUSE_N with the normaliser being equal to 1.
>
> > l.57 "Our unsupervised feature extraction method is extremely powerful"
>
> We believe the method has the potential to be extremely powerful, particularly when applied to large-scale and high-dimensional data, but we will clarify that further evidence is needed to demonstrate this conclusively.
>
> > l.108f: Kübler et al show [...]
>
> Thank you for pointing this out, we will cite this specific result.

---

> > ### Comment · Reviewer_fFeS · 2023-08-15
> > **Thanks for the rebuttal**
> >
> > Thank you for the detailed rebuttal. In particular regarding the motivation of MMD_N. I want to strongly encourage you to bring this motivation at least in parts to the main paper.
> >
> > Thanks also for providing the time comparison, that is indeed good to know, I somewhat thought the kernel compuation was the bottleneck.
> >
> > I have further **Question** left regarding MMDAgg:
> > - My take from your experiments and analysis is that your method powerwise performs similarly to MMDAgg, but is twice as fast. This is nice but not groundbreaking I would argue.
> >   - But you say that when considering more and more kernels then your method has an advantage, because of the multiple corrections in MMDAgg. I don't quite see this from the theory. Is it possible to provide an experiment that showcases this? Overall it would be nice to have some setting where you show that MMDAgg has some power-problems.
> >
> >
> > [sorry for being a bit late]

---

> > > ### Author Response · Authors · 2023-08-18
> > >
> > > Thank you for engaging in the rebuttal. We will definitely be incorporating the additional motivation arguments in the final version of the paper.
> > >
> > > Let us bring some clarifications regarding the following sentences from our "Additional Motivation for our proposed statistics and work" rebuttal related to the theoretical power of MMD-FUSE and of tests based on multiple testing in the regime with an increasing number of kernels/sub-tests.
> > >
> > > > This can be beneficial in the case where the number of kernels considered increases (is large). In that setting, the multiple testing approach MMDAgg combines more and more kernels, the level needs to be corrected differently depending on the number of kernels, and it is not theoretically guaranteed how the power of MMDAgg behaves in that setting (i.e. it could maybe go to zero).
> > >
> > > When performing multiple testing, in order to correctly control the type I error at level $\alpha$, the level of the $T$ sub-tests need to be corrected.
> > >
> > > The simplest method for that is the Bonferroni correction which guarantees that the overall test has level $\alpha$ if the $T$ sub-tests are each run with level $\alpha/T$. Loosely speaking, if the number of sub-tests $T$ becomes very large, every sub-test needs to be run with very small level which means that the sub-tests are very conservative (less likely to reject), so the overall test has low power. In fact, for the Bonferroni correction, it can be theoretically proved that the power goes to zero in the limit of $T$ tending to infinity. This is because the tests reject if the statistic (e.g. MMD) is strictly greater than the $(1- \alpha/T)$-quantile, but in the limit this becomes the $1$-quantile which is essentially the maximum, so the sub-tests would never reject.
> > >
> > > Instead of relying on a Bonferroni correction, the authors of MMDAgg propose using $u/T$ where $u$ is chosen to be the largest value (using data) such that the overall test has estimated level $\alpha$. It is clear that $u$ is larger than $\alpha$, hence their method is more powerful than performing a Bonferroni correction. However, the authors do not theoretically study the behaviour of the MMDAgg test in the limit where the number of sub-tests $T$ (each with a different kernel) tends to infinity. The power of the test in that setting could, as in the Bonferroni case, simply go to zero.
> > >
> > > As we mentioned in the rebuttal (see quote below), our theoretical study not only covers the case where the number of kernels (analogous to $T$ above) increases to infinity, but FUSE is actually well-defined in the limit where the average becomes a distribution over kernels, which is clearly not the case of any test relying on multiple testing. This is a real advantage of our proposed method compared to others such as MMDAgg.
> > >
> > > > We do not suffer from such issues with FUSE as even when increasing the number of kernels we still have only one statistic and quantile. The FUSE statistic is even defined in the limit of an infinite number of kernels (continuous/uncountable collection of kernels) by considering distributions on the space of kernels, which is clearly not the case for MMDAgg (only defined for discrete collection of kernels). Moreover, while the MMDAgg approach is only useful for hypothesis testing, having a quantity like FUSE combining multiple kernel-based measures of distance with exponential concentration bounds could be of interest in a wide range of applications.
> > >
> > > As mentioned above, the *theoretical* power of MMDAgg has not been studied by the authors in the setting of increasing $T$, and it is unclear what the limiting behaviour would be. However, we do point out that the authors consider this setting *empirically* and observe in their experiments (MMDAgg Section 5.7) that MMDAgg retains power when combining kernels with multiple bandwidths. For this reason, it is challenging to provide an "experiment that showcases" the power advantage when considering "more and more kernels".
> > >
> > > We argue that our experiment distinguishing between CIFAR-10 and CIFAR-10.1 in Appendix A.5 provides a setting where MMD-FUSE achieves significantly higher power than MMDAgg while being twice as fast. We also emphasise that, unlike our proposed MMD-FUSE test, the MMDAgg test has not been proven to achieve the optimal MMD separation rate, and MMD-FUSE also has the advantage of being a single statistic.

---

> > > > ### Comment · Reviewer_fFeS · 2023-08-18
> > > > **Reply**
> > > >
> > > > Thank you for the additional clarification.
> > > >
> > > > Let me rephrase my understanding regarding MMD Fuse vs MMDAgg:
> > > > - The authors of the present paper show beneficial behavior for the case of considering many kernels.
> > > > - For MMDAgg this limiting behavior is neither theoretically nor empirically well-studied.
> > > >
> > > > The author do also not provide experiments on this.
> > > >
> > > > Ergo:
> > > > - MMDFuse has a more complete theory than MMDAgg
> > > > - Whether it performs better when considering many kernels **is not known**
> > > >
> > > >
> > > > > We argue that our experiment distinguishing between CIFAR-10 and CIFAR-10.1 in Appendix A.5 provides a setting where MMD-FUSE achieves significantly higher power than MMDAgg while being twice as fast.
> > > >
> > > > Sure, on this setting MMDFuse is better. But on other settings MMDAgg is better by similar margins (example Figure 1 bottom right). So no general winner. Also this setting does also not address the question of using more kernels.
> > > >
> > > >
> > > > ---
> > > > In conclusion my judgement remains the same, because the shown improvements over MMDAgg are mainly a factor 2 in runtime.:
> > > >
> > > > 6: Weak Accept: Technically solid, **moderate-to-high impact** paper, with no major concerns with respect to evaluation, resources, reproducibility, ethical considerations.

---

> > > > > ### Author Response · Authors · 2023-08-21
> > > > >
> > > > > As requested, we have run an additional experiment increasing the number of kernels for MMD-FUSE and for MMDAgg from 10 to 1000 kernels. We consider the problem of distinguishing between a uniform distribution and a perturbed uniform distribution, having 500 samples from each. For both tests, both Gaussian and Laplace kernels are used with 5 to 500 bandwidths each (the bandwidths correspond to a finer discretisation of the collection l.669). We have updated the anonymised repository (the link can be found l.655) with the reproducible code for this experiment in extra_experiment.ipynb, further details about the experiment can be found there.
> > > > >
> > > > > The power results averaged over 200 repetitions are the following (we are not able to attach a plot in OpenReview sorry):
> > > > >
> > > > > - number of kernels : [10,  50,  100,  200,  400,  600,  800,  1000]
> > > > > - power of MMD-FUSE: [0.94,  0.905,  0.9,  0.91999996,  0.91999996,  0.93,  0.93,  0.91499996]
> > > > > - power of  MMDAgg: [0.745,  0.72999996,  0.74,  0.73499995,  0.715,  0.71999997,  0.73499995,  0.74]
> > > > >
> > > > > First, we observe that MMD-FUSE achieves significantly higher power than MMDAgg in this setting. Second, we observe that both MMD-FUSE and MMDAgg retain their power when increasing the number of bandwidths. As previously mentioned, this matches the observations of the authors of MMDAgg presented in their Section 5.7, the intuition behind it being that a discretisation of 10 bandwidths is enough to capture all the information in this problem and that a finer discretisation does not help.
> > > > >
> > > > > > The authors of the present paper show beneficial behaviour for the case of considering many kernels.
> > > > >
> > > > > Yes, thank you for recognising this.
> > > > >
> > > > > > For MMDAgg this limiting behaviour is neither theoretically nor empirically well-studied.
> > > > >
> > > > > The point we are making is that MMD-FUSE is theoretically well-defined even in the limit while the limiting behaviour of MMDAgg has not been theoretically studied by the authors (though they have studied it empirically in their Section 5.7).
> > > > >
> > > > > > The author do also not provide experiments on this.
> > > > >
> > > > > We have now provided the results and reproducible code for the requested experiment.
> > > > >
> > > > > > MMDFuse has a more complete theory than MMDAgg
> > > > >
> > > > > We are happy to hear you are satisfied with the theoretical completeness of our paper.
> > > > >
> > > > > > Whether it performs better when considering many kernels is not known
> > > > >
> > > > > We do not observe power deterioration when considering many more kernels.
> > > > >
> > > > > —
> > > > >
> > > > > We appreciate your engagement during this discussion period and hope we have addressed your concerns in our replies and with the additional experiment. If so, we ask if you would consider increasing your score.

---

> > > > > > ### Comment · Reviewer_fFeS · 2023-08-22
> > > > > > **Thanks for the experiments**
> > > > > >
> > > > > > Thank you very much for the additional experiments (I was actually not aware that you were planning to add those). I think adding these in would nicely show that neither MMDAgg nor MMDFuse (empirically) suffer a power loss from fine-graining the kernels. Of course it becomes unnecessarily expensive. For MMDFuse you provide theoretical results that this holds true, whilst for MMDAgg this is not known to be guaranteed.
> > > > > >
> > > > > > Overall I will increase my score from 6 to 7. I think the authors have sharpened their comparison to MMDAgg with the additional experiments. This should be helpful to readers to clearly see the merits.

---

### Official Review · Reviewer_jkkz · 2023-06-30

**Soundness:** 4 excellent
**Presentation:** 4 excellent
**Contribution:** 4 excellent
**Rating:** 8
**Confidence:** 2

**Summary:**

This paper looks into the problem of permutation based kernel two-sample testing. The paper is motivated by two main challenges:

1. The first one being the fact that selecting kernel hyper-parameters through standard data splitting often reduces the statistical power due to using less data than we actually have.
2. The second challenge being that if we don't select a data specific hyperparameter for the test, a poor kernel choice will also lead to a decrease in statistical power.

To tackle these problems, the authors proposed the MMD-FUSE method that can select kernels adapted to the data without data splitting. They first restate a result from Hemerik and Goeman Theorem 2 to justify/emphasise that any permutation invariant representation/function of the data can be used a statistic that have controlled type 1 error rate. In addition justifying the adequacy of median heuristics.

The second contribution is that they proposed to combine multiple MMD via a set of kernels, and as such optimise the test power.


**Strengths:**

The problem is well motivated as the trade off between data splitting to learn optimal kernel v.s. not splitting the data but selecting some heuristic kernel has been a long standing problem in the community. This authors explained their arguments very clearly. In addition, not only did they justify their method by showcasing the optimal MMD separation rate (Typo in line 290 btw), I appreciate they included a discussion with computationally faster sub-optimal tests to highlight between the trade off of test power and computational efficiency.

**Weaknesses:**

The authors have discussed the weakness in their conclusion section so I don't think I have much to add. There is some recent work on not using permutation test at all to do two sample test, such as the work of Shekhar et al. 2022. I understand it is not necessary to compare the two types of approach but might be also good to mention non-permutation based approach as a side note to complete your already-detailed background section on testing?


---

Shekhar et al. 2022 "A Permutation-free Kernel Two-Sample Test" arXiv:2211.14908

**Questions:**

Although you have this prior over a set of kernels which this set could potentially be infinite, in practice you would sample from this prior to compute the test statistic right? How would your approach compare to standard Bongerroni correction technique or its extension weighted bonferroni correction? Would be interesting to hear about your comments on this.

**Limitations:**

No.

---

> ### Author Rebuttal · Authors · 2023-08-08
>
> *[Comment to all reviewers (Author Rebuttal) posted above]*
>
> Thank you for your generous praise, and comments. We will incorporate answers to the following into our revised work.
>
> &nbsp;
>
> > There is some recent work on not using permutation test at all to do two sample test, such as the work of Shekhar et al. 2022. I understand it is not necessary to compare the two types of approach but might be also good to mention non-permutation based approach as a side note to complete your already-detailed background section on testing?
>
> We briefly mentioned this work in our Background section on line 157 as part of the 'studentised asymptotic tests' as opposed to ‘permutation-based non-asymptotic tests’. We will expand our discussion of this interesting alternative to constructing a two-sample test with advantages (still quadratic but computationally more efficient as no permutation required) and disadvantages (inevitable loss of power due to the fact that the kernel matrix cannot be used entirely, otherwise asymptotic normality is broken), which is the usual tradeoff between power and computational efficiency. This could also be a topic for investigation in future work.
>
> &nbsp;
>
> > Although you have this prior over a set of kernels which this set could potentially be infinite, in practice you would sample from this prior to compute the test statistic right?
>
> There are three possible settings.
>
> Firstly, the prior can have support over a finite set of kernels, in which case the expectation can easily be evaluated as an average. This is the case in our experiments which directly compare against other similar tests (e.g. MMDAgg).
>
> Secondly, in some cases the prior can have support over an infinite set of kernels but the expectation might still admit a known closed form which can easily be evaluated. This is for example the case when using FUSE1 with a Gaussian kernel and a Gamma prior on its bandwidth parameter, as shown in Appendix B the resulting expectation (i.e. the mean kernel) is the commonly-used rational quadratic kernel which can be implemented in practice.
>
> Thirdly, the prior can have support over an infinite set of kernels and the expectation cannot be expressed in a known closed form. As mentioned by the reviewer, in this case, it would certainly be sensible to sample from the prior to produce a new prior with finite support, for which the expectation becomes a simple weighted average. This would still be very different from performing multiple testing with a (weighted) Bonferroni correction of the test levels (we get a different quantile for each kernel), since we would still be using a 'soft maximum' via FUSE and thus be avoiding multiple testing (we have a single quantile for our FUSE statistic which combines many MMD values with different kernels). Note that empirically FUSE would greatly outperform the Bonferroni multiple testing approach, as it matches the performance of the multiple testing approach MMDAgg which itself greatly outperforms the Bonferroni multiple testing correction (both theoretically and empirically).
>
>
> > How would your approach compare to standard Bonferroni correction technique or its extension weighted Bonferroni correction? Would be interesting to hear about your comments on this.
>
> Standard Bonferroni correction is needed when we do multiple testing with different kernels. However, this leads to a term of $O(\sqrt{\log(|K|)})$ in the power.
> In the worst case for our method, where only a single kernel gives rise to a large normalised MMD and the rest give approximately zero normalised MMD, we would incur such a penalty; but in the case that several kernels do so the penalty would be considerably smaller. This is what is empirically observed. Further, this makes testing with very large or infinite sets of kernels possible (though not always computationally practical), since otherwise $|K| \to \infty$ leads to zero power.

---

> > ### Comment · Reviewer_jkkz · 2023-08-12
> > **Thanks for the rebuttal**
> >
> > After reading the rebuttal, I have no further questions. Thank you.

---

> > > ### Author Response · Authors · 2023-08-21
> > >
> > > Glad to hear all your questions have been answered. Thank you again for the very supportive comments!

---

### Official Review · Reviewer_xB7K · 2023-07-04

**Soundness:** 3 good
**Presentation:** 3 good
**Contribution:** 2 fair
**Rating:** 7
**Confidence:** 5

**Summary:**

The paper studies the question of choice of critical values for the MMD test. The authors introduce a new MMD-based test that allows data-driven critical values based on permutation test, without the need to train a new kernel for every permutation. The main advantage of this new method is that it does not require sample splitting and the whole sample can be utilized for the computation of the test value.

pros:
- Experiments show improvements in some cases compared to sample-splitting methods.
- The authors provide plenty of theoretical results in the appendix, including concentration results for the test both under null and alternative.

cons:
- The analysis is very conservative. At no point we can state that the distribution of the permuted test conditional on the observed data approximates that of the original test. Theorem 1 in particular is an upper bound, so there is no guarantees that the type I error of the test will be close to the desired level alpha. It is not clear how crude this upper bound is. I suggest this limitation should be emphasised in the text below.
- I think theoretical analysis can be simplified. For example, proof of Theorem 6 does not have to reproduce Vershynin and Rudelson's proof, it is sufficient to integrate their bound. Also, it is not clear to me why it is necessary to include a proof of Bounded Differences Inequality in the appendix. Instead, one can refer to Lugosi et al Concentration Inequalities, Theorem 6.7.
- There is no dependence of prior on the data in the experiments, please correct me if I'm wrong.

other comments:

line 174 m+ m -> n + m

line 222 inconsistent notation MMD_k or MMD(.; k)

line 829, display below: not sure why t appeared in second line and why it disappeared in the third line

line 747 FUSE depends on lambda, what depends on t then?

line 872 sup_y -> sup_rho?

line 873 what is N?

Serious concerns:

- Section E.3 proof of Theorem 2 (which in my understanding one of the main theoretical contributions of the paper). Could the authors please elaborate on what happens below line 926 in detail? The sup over rho disappears in line 2. The "object" S_rho appears to be random, depending on the observed data Z, and it appears in the final bound thanks to the Markov inequality.
- I can't find proof of Theorem 3 in the appendix with Ctrl + F

To my best understanding if there are mistakes above section E.3 they should be fixable.

### My rating is conditional on whether the authors can fix these two serious concerns. Otherwise, the paper must be rejected.


**Strengths:**

-

**Weaknesses:**

-

**Questions:**

-

**Limitations:**

-

---

> ### Author Rebuttal · Authors · 2023-08-08
>
> *[Comment to all reviewers (Author Rebuttal) posted above]*
>
> Thank you for your thorough feedback, positive evaluation of our work, and for pointing out ways in which we can more clearly communicate our contribution. We also thank you for pointing out the small error in the proof of Th 2 due to incorrect ordering of steps, the fix to which we give below.
>
> > The analysis [...]
>
> The statement of Th 1 which states only that the level is smaller than $\alpha$ appears to be conservative, but in fact we can make stronger statements about the test arising from this theorem. Specifically, assuming distinct data points, the level can be shown to be **exactly** equal to $\lfloor (B +1) \alpha\rfloor / (B +1)$ (this result follows from Prop 2 of H&G). Thus the bound is extremely tight, and in many cases the equality is achieved (e.g. if $\alpha=0.05$ and $B+1$ is a multiple of $\alpha^{-1}=20$). Even the assumption of distinct data points can be relaxed (Proposition 3 of Hemerik and Goeman) using a tie-breaking strategy (randomly choosing an ordering of the equal points) while preserving the exact level $\lfloor (B +1) \alpha\rfloor / (B +1)$. This is an important point which we will add to the paper.
>
> > Th 6 V&R
>
> Th 6 is actually derived from an intermediate result (not a full theorem) in V&R's paper, which they do not derive with explicit constants. In order to provide the concentration results (Th 5) with the (surprisingly small) constants, we need these, so we reproduce this part of their proof with explicit constants.
>
> > Bounded Diff. Ineq. Lugosi
>
> The “bounded difference lemma” (Th 11) we give is not the same as the bounded difference inequality, though it is closely related. In fact, our result could be used as an intermediate step in proving the latter, but we note that this would not be the usual method (e.g. Lugosi et al. use an “entropy” method instead).
>
> The usual bounded difference inequality does **not** give the concentration we need, since the FUSE statistics do not have the bounded difference property, only $\widehat{\mathrm{MMD}}^2$ does. Our “bounded difference lemma” is an MGF bound and leads to the $\widehat{\mathrm{MMD}}^2$ MGF bound of Theorem 12. This is applied to prove Theorems 9 and 10, since $\widehat{\mathrm{FUSE}}_1$ takes the approximate form of an MGF. The fact that we have to prove a variant form of an existing concentration inequality to get concentration for our logsumexp statistics is similar to how the same is required in PAC-Bayesian proofs, where e.g. PAC-Bayes Bernstein inequalities also require modified proof techniques that mirror those used to prove the usual Bernstein inequalities.
>
> > prior dependence
>
> The prior used in the experiments uses data in a permutation-independent way to choose bandwidths for a fixed set of kernels (cf l.317-324) as it uses the set of inter-sample distances $\|z-z'\|$ for $z,z'\in Z$ (l.320) for bandwidth selection. The prior is a uniform distribution over a support chosen using the data.
>
> > l.174, 222, 829, 872
>
> We thank the reviewer for pointing these typos out, which are now fixed throughout the paper.
>
> > l.747
>
> We will clarify this in the final version. FUSE depends on $\lambda$ and the same $\lambda$ appears on the RHS of the equation l.745. For the bound on -FUSE of l.746, FUSE depends on $\lambda$ but the RHS holds for any $t$ in the range specified on l.747, in particular the parameter $t$ can be optimised to match the orders of the two terms on the RHS which is not possible for the bound on l.745.
>
> > l.873
>
> We will fix this, here 1/N = 1/n + 1/m.
>
> > Section E.3 [...]
>
> Thank you for pointing this out, you are correct, as currently written $S_\rho$ depends on Z, when it should not. The problem here is that as presented, the bound from Theorem 5 (l. 929) is used **after** Markov, where instead it should be used **before**. This can easily be fixed by redefining $S_\rho$, replacing this quantile term by a data-independent upper bound on it. The same issue appears with the same fix in the proof of Th 3.
>
> For clarity, we present the details of this modified proof below along with additional details to clarify what is happening below 926:
>
> From Th 5 with $\lambda = cn / \kappa$, we have
> $\mathrm{quantile}\ \widehat{\mathrm{FUSE}}(\sigma Z) \leq C_1 \kappa (1+\log(1/\alpha))/n$
> for some positive constant $C_1$.
> We now define $S_\rho$ using this quantile upper bound, so that $S_\rho$ is not data-dependent, we let
> $$
> S_\rho := \mathrm{MMD}^2(p,q;K_\rho) - \mathrm{KL}(\rho,\pi) / \lambda - C_1 \kappa (1+\log(1/\alpha))/n.
> $$
>
> From l.926 onwards:
>
> Under the alternative and using Th 5,  we have
> $$
> \mathbb{P}(\widehat{\mathrm{FUSE}}(Z) \leq \mathrm{quantile}\ \widehat{\mathrm{FUSE}}(\sigma Z))
> \ \leq\
> \mathbb{P}(\widehat{\mathrm{FUSE}}(Z) \leq C_1 \kappa (1+\log(1/\alpha))/n)
> $$
>
> Now, we add $S_\rho$ to both sides and cancel out the quantile upper bound to obtain
>
> $$
> \mathbb{P}(\widehat{\mathrm{FUSE}}(Z) \leq C_1 \kappa (1+\log(1/\alpha))/n) =
> \mathbb{P}( \mathrm{MMD}^2(p,q;K_\rho) - \mathrm{KL}(\rho,\pi) / \lambda - \widehat{\mathrm{FUSE}}(Z)\geq S_\rho).
> $$
>
> Continuing from the second line after l.926 the reasoning continues as given below:
>
> The third line of the equations l.926 holds by definition of $\widehat{\mathrm{FUSE}}_1(Z)$ (note that there is a typo and the MMD term in the supremum over $\rho'$ should depend on $\rho'$ rather than on $\rho$).
>
> The fourth line holds since the supremum over all $\rho'$ is greater than when it is evaluated at the specific $\rho$.
> We note there is no supremum over $\rho$ since the above applies for any fixed $\rho$.
>
> The fifth line (which should be an inequality) holds by Markov/Chebyshev's inequality (valid since $S_\rho$ is now data-independent), and the last line holds by Th 14.
>
> The rest of the proof is completed as given.
>
> > can't find proof Th 3
>
> The proof of Th 3 is presented in Appendix E.4 directly after the proof of Theorem 2. We will rename this section more appropriately.

---

> > ### Comment · Reviewer_xB7K · 2023-08-11
> >
> > Thank you for the answers. Could you please write here the modified lines 926 - 935 as a comment? It would help a lot to verify this.
> >
> > - Th 6 V&R. If you insist on having smaller constants that's fine, but I assume the constants are not important for your proof, and in case you actually have improvements for the constants, it should be a matter for a separate paper dedicated for the constants of H-W inequality.
> >
> > - Bounded Diff. Ineq. Lugosi;  I understand that you version of bouded differences is stronger than classical form, but the version from the book is even stronger.

---

> > > ### Author Response · Authors · 2023-08-12
> > >
> > > > Thank you for the answers. Could you please write here the modified lines 926 - 935 as a comment? It would help a lot to verify this.
> > >
> > > To summarise, only the definition of $S_\rho$, and lines 926-929 change. Running through the proof on line 926 gives the same result on 928, with the altered definition of $S_\rho$.
> > >
> > > However, we give a full proof (without align since Markdown won't handle it) below for understanding:
> > >
> > > &nbsp;
> > >
> > > For any $\rho$, we (re)-define
> > > $$  S_{\rho} = MMD^2(p, q; K_{\rho}) - \frac{KL(\rho, \pi)}{\lambda} - \frac{C \kappa (1 + \log \alpha^{-1})}{n} $$
> > >
> > > [...]
> > >
> > > The Type II error under the full permutation set can be bounded as
> > >
> > > $$ \Pr_{p \times q}\left( \widehat{\operatorname{FUSE}}(Z) \le \text{quantile}_{1-\alpha_B, \mathcal{G}} \widehat{\operatorname{FUSE}}(\sigma Z) \right) $$
> > >
> > > $$ \le \Pr_{p \times q}\left( \widehat{\operatorname{FUSE}}(Z) \le \frac{C_2 \kappa (1 + \log \alpha^{-1})}{n} \right)$$
> > >
> > > since by Theorem 5 with $\lambda = cn / \kappa$,
> > > $$ \text{quantile}_{1-\alpha_B, \mathcal{G}} \widehat{\operatorname{FUSE}}(\sigma Z) \le \frac{C_2 \kappa (1 + \log \alpha^{-1})}{n}. $$
> > >
> > > Now
> > > $$ \Pr_{p \times q}\left( \widehat{\operatorname{FUSE}}(Z) \le \frac{C_2 \kappa (1 + \log \alpha^{-1})}{n} \right)$$
> > >
> > > $$ = \Pr \left(MMD^2(p, q; K_{\rho}) - \frac{1}{\lambda}KL(\rho, \pi) - \widehat{\operatorname{FUSE}}(Z) \ge S_{\rho} \right)$$
> > > simply by rearranging terms and introducing the definition of $S_\rho$.
> > >
> > > The next steps are by $\sup f(\rho') \ge f(\rho)$, Chebyshev's, and Theorem 14:
> > > $$ \Pr \left(MMD^2(p, q; K_{\rho}) - \frac{1}{\lambda}KL(\rho, \pi) - \widehat{\operatorname{FUSE}}(Z) \ge S_{\rho} \right)$$
> > >
> > > $$ = \Pr \left(MMD^2(p, q; K_{\rho}) - \frac{1}{\lambda}KL(\rho, \pi) - \sup_{\rho'} \left( \widehat{MMD}^2(Z; K_{\rho'}) - \frac{1}{\lambda}KL(\rho', \pi)  \right) \ge S_{\rho} \right) $$
> > >
> > > $$\le \Pr \left(MMD^2(p, q; K_{\rho}) - \widehat{MMD}^2(Z; K_\rho) \ge S_{\rho} \right) $$
> > >
> > > $$\le \frac{1}{S_{\rho}^2} \mathbb{V} \left[\widehat{MMD}^2(Z; K_{\rho}) \right]  $$
> > >
> > > $$\le \frac{C_1}{S_{\rho}^2} \left( \frac{4\kappa MMD^{2}}{n} + \frac{\kappa^2}{n^2} \right). $$
> > >
> > >
> > > This term is upper bounded by $\beta / 2$ if we set
> > > $$
> > >   S_{\rho}^2 > \frac{2C_1}{\beta}\left( \frac{4\kappa MMD^{2}}{n} + \frac{\kappa^2}{n^2} \right).
> > > $$
> > >
> > > [Thus, the type II error rate will be controlled if l.924 is true]
> > > [with this new definition of $S_\rho$, the proof continues in exactly the same way, skipping line 929 which we used at the start]
> > >
> > >
> > >
> > >
> > >
> > > ---
> > >
> > >
> > > > Th 6 V&R. If you insist on having smaller constants that's fine, but I assume the constants are not important for your proof, and in case you actually have improvements for the constants, it should be a matter for a separate paper dedicated for the constants of H-W inequality.
> > >
> > > To be clear, V+R do not give numerical constants at all (only their existence is proved), which we need for our statements of Theorem 5.
> > > We do not intend to claim this as a contribution, the proof is only included so that we are not stating the numerical constants without justification.
> > > We provide the numerical constants in Theorem 5 even though they are not strictly necessary, as we hope they could be useful when adapting FUSE statistics to other settings.
> > > In particular, the constants help understand for which values of  $\lambda$ the bounds of Theorem 5 hold, and also allows to understand the relation between the two RHS terms which can be matched by tuning $\lambda$.
> > >
> > >
> > >
> > >
> > > ---
> > >
> > > > Bounded Diff. Ineq. Lugosi; I understand that you version of bounded differences is stronger than classical form, but the version from the book is even stronger.
> > >
> > > We agree that Lugosi et al. is stronger than the classical form of bounded differences, however, we do not believe that it can straight-forwardly be used to prove Theorem 9, which is *not* a concentration bound.
> > > Theorem 9 instead relates the expectation $\mathbb{E} \widehat{\operatorname{FUSE}}_1$ to $\operatorname{FUSE}_1$ (l. 845), an expression involving the true MMD.
> > > There is no in high probability statement here, these quantities are deterministic, and require Theorems 11+12 to prove, rather than any concentration result.
> > >
> > > It *might* be possible to use Lugosi et al.'s result to prove concentration for $\widehat{\operatorname{FUSE}}_1$ around its mean instead, a slightly different but still interesting result that we can also show by combining Theorems 9 + 10.
> > > However, setting $f(Z) = \widehat{\operatorname{FUSE}}_1(Z)$ in their result, we still need a bound for
> > > $$ \sum_i (f(Z) - f(Z_i)^2 = \sum_i (\log \frac{E \exp(\lambda \widehat{MMD}^2(Z))}{E \exp(\lambda \widehat{MMD}^2(Z_i))})^2.$$
> > > It is not obvious to us that a bound will follow that is any better than $O(1)$ in the sample size, while we need $O(1/n)$ to match our theorems.
> > > We would argue that even if it turns out this is possible, it is simpler just to work using Theorems 11+12, which have short and simple proofs; however, we are happy to discuss further if you have an idea how this could be done.

---

> > > > ### Comment · Reviewer_xB7K · 2023-08-15
> > > >
> > > > Thank you for corrections, that version seems to be correct. Thanks.
> > > >
> > > > > We will clarify this in the final version. FUSE depends on  and the same  appears on the RHS of the equation l.745. For the bound on -FUSE of l.746, FUSE depends on  but the RHS holds for any  in the range specified on l.747, in particular the parameter  can be optimised to match the orders of the two terms on the RHS which is not possible for the bound on l.745.
> > > >
> > > > Sorry for coming back to this, but it just does not compute in my head why the lower bound depends on some $t$ instead of $\lambda$. You could as well optimise it wr.t. $t$ right away.
> > > >
> > > > Furthermore, in the proof of lower bound of Theorem 5 you write
> > > >
> > > > > l 834 for any function ... and $s > 0$
> > > >
> > > > Then you set
> > > >
> > > > > Combining this with the above results and $t = -s$
> > > >
> > > > In the conditions of the Theorem $t > 0$ and that means $s < 0$. Furthermore, you claim that the bound in the display above l830 holds for $|t| < ... $. Are you sure it holds for any negative $t$? Could you please double check the signs in the first inequality. My concern is that, although I am not very much familiar with PAC-Bayesian bounds, I thought that they don't usually guarantee any lower tails. Correct me if I'm wrong.

---

> > > > > ### Author Response · Authors · 2023-08-16
> > > > >
> > > > > Thank you for engaging in the discussion. We agree that the current "Lower Bound" section  (l.834 - l.837) of the proof of Theorem 5 is slightly confusing, even without the contradicting statements about $s$. We present a clearer reasoning below, which we hope address your concerns, and will also update the final version of the paper accordingly.
> > > > >
> > > > > > You could as well optimise it w.r.t. $t$ right away.
> > > > >
> > > > > We agree that this is probably simpler and have added an optimisation step below (see final lines of this comment).
> > > > >
> > > > > > In the conditions of the Theorem $t>0$ [...]
> > > > > > Furthermore, you claim that the bound in the display above l830 holds for $|t|<\dots$ . Are you sure it holds for any negative $t$?
> > > > > > Could you please double check the signs in the first inequality.
> > > > >
> > > > > The reasoning l.829-830 indeed holds for any $|t|< (4\sqrt{2} \sup_k \bar{U}_k)^{-1}$ (by Theorem 7). The upper bound of Theorem 5 corresponds to the result l.829-830 with $t = \lambda > 0$, divided through by $\lambda$.
> > > > > We note that although the signs in the first inequality of l.829-830 are correct, there is a typo and that the leading term $1/t$ should not appear. The rest of ll.829-830 is correct.
> > > > >
> > > > > > My concern is that, although I am not very much familiar with PAC-Bayesian bounds, I thought that they don't usually guarantee any lower tails.
> > > > >
> > > > > PAC-Bayesian bounds are often not simple and symmetric two-sided bounds on $|X - EX|$, but these do exist (e.g. Alquer, Ridgway, Chopin, 2016, On the properties of variational approximations of Gibbs posteriors). More typical is what we see here, that the upper and lower tails take somewhat different forms, e.g. the famous “small-kl” PAC-Bayes bound (Langford & Seeger, 2003; Maurer, 2004) is two-sided but asymmetric, so the upper and lower tails have slightly different rates. The literature is certainly more interested in the upper bounds, but the proof techniques apply similarly.
> > > > >
> > > > > **Revised "Lower Bound" section  (l.834 - l.837)**
> > > > >
> > > > > For the lower tails, we note that for any function $f(Z, k)$ and $\lambda > 0, s > 0$,
> > > > > $$
> > > > > \frac{1}{\lambda}\log E_{k \sim \pi(\langle Z \rangle)} \exp(\lambda f(Z, k))
> > > > > $$
> > > > > $$
> > > > > \ge E_{k \sim \pi(\langle Z \rangle)}[f(Z, k)]
> > > > > $$
> > > > > $$
> > > > > = -\frac{1}{s} E_{k \sim \pi(\langle Z \rangle)} [\log (\exp \left( -s f(Z, k) \right))]
> > > > > $$
> > > > > $$
> > > > > \geq -\frac{1}{s} \log E_{k \sim \pi(\langle Z \rangle)} [\exp \left( -s f(Z, k) \right)]
> > > > > $$
> > > > >
> > > > > where both inequalities are by Jensen's inequality, which gives
> > > > > $$
> > > > > -\frac{1}{\lambda}\log E_{k \sim \pi(\langle Z \rangle)} \exp(\lambda f(Z, k)) \leq \frac{1}{s} \log E_{k \sim \pi(\langle Z \rangle)} [\exp \left( -s f(Z, k) \right)].
> > > > > $$
> > > > > for $s > 0$.
> > > > >
> > > > > Using this result with the appropriate function $f$ for $\widehat{\mathrm{FUSE}}^1$ and $\widehat{\mathrm{FUSE}}^N$, together with the bound l.829 and l.831 (over the defined ranges) applied to the log-expectation term with $ t=-s $ (which is negative and satisfies the condition $ |t|< (4\sqrt{2} \sup_k \bar{U}^k)^{-1}$), we find that
> > > > > $$
> > > > > -\widehat{\mathrm{FUSE}}^1(Z) \le_{1-\delta} \frac{4\kappa^2 s}{n(n-1)} + \frac{\log\frac{1}{\delta}}{s},
> > > > > $$
> > > > > $$
> > > > > -\widehat{\mathrm{FUSE}}^N(Z) \le_{1-\delta} \frac{16 s}{n(n-1)} + \frac{\log\frac{1}{\delta}}{s}.
> > > > > $$
> > > > >
> > > > > Unlike the upper bounds, where rates depend on setting $\lambda \asymp n$, we can freely choose $s$ here within the specified range to optimise the lower bounds.
> > > > >
> > > > > A simple further bound on these follows by setting $s$ to its extremal values, giving:
> > > > > $$
> > > > > -\widehat{\mathrm{FUSE}}^1(Z) \le_{1-\delta} \frac{\sqrt{2} \kappa (\frac{1}{4} + 8\log\frac{1}{\delta})}{\sqrt{n(n-1)}},
> > > > > $$
> > > > > $$
> > > > > -\widehat{\mathrm{FUSE}}^N(Z) \le_{1-\delta} \frac{\sqrt{2} ( \frac{1}{2} + 16\log\frac{1}{\delta})}{\sqrt{n(n-1)}}.
> > > > > $$

---

> > > > > > ### Comment · Reviewer_xB7K · 2023-08-18
> > > > > >
> > > > > > Once again, your lower bounded the moment generating function for negative $\lambda$, while it is required to be upper-bounded. Just a quick common sense test: a lower bound for $f(Z)$ is an upper bound for $-f(Z)$. For an upper bound on $-f(Z)$ you need to have an upper bound on $ E \exp( - \lambda f(Z))$ for $\lambda > 0$. I only see a lower bound on $ E \exp( - \lambda f(Z))$ in your answer above. I really doubt the Jensen trick alone can give you required bound.

---

> > > > > > > ### Author Response · Authors · 2023-08-18
> > > > > > >
> > > > > > > Apologies, we are having difficulty understanding your question - which exact step of our proof do you have trouble with? We have written it out in greater detail below so that if you have any specific issue it will be easier to see.
> > > > > > >
> > > > > > > $\lambda$ is a **fixed non-negative parameter** of FUSE rather than a dummy variable so we are not sure where we set it negative.
> > > > > > > We wonder if there is confusion about the term $f(Z)$: we are not writing $f(Z) = FUSE_1$, we are writing $f(Z) = MMD(Z)$. With first the setting $f(Z) = FUSE_1$ it is true we would need to upper bound  $E \exp(-s f(Z))$ for positive $s$, but we are considering the second. This makes our proof somewhat different in formulation from usual concentration proofs.
> > > > > > >
> > > > > > > ### Highly detailed proof
> > > > > > > For $FUSE_1$ with parameter $\lambda > 0$, we have
> > > > > > > $$ -FUSE_1 = -\frac{1}{\lambda}\log E_{k \sim \pi(\langle Z \rangle)} \exp(\lambda MMD(Z, k)) $$
> > > > > > > Now apply Jensen with $E\exp(X) \ge \exp(EX)$ to the above, noting that the inequality is reversed due to the negative sign and $\lambda > 0$, and that the log and exp terms cancel:
> > > > > > > $$ -\frac{1}{\lambda}\log E_{k \sim \pi(\langle Z \rangle)} \exp(\lambda MMD(Z, k)) \le - E_{k \sim \pi(\langle Z \rangle)}  MMD(Z, k).$$
> > > > > > > Next introduce the dummy variable $s > 0$ and see that
> > > > > > > $$ - E_{k \sim \pi(\langle Z \rangle)}  MMD(Z, k) = \frac{1}{s} \times (- s E_{k \sim \pi(\langle Z \rangle)} MMD(Z, k)) = \frac{1}{s} \log \circ \exp (- s E_{k \sim \pi(\langle Z \rangle)} MMD(Z, k)). $$
> > > > > > > Since $\frac{1}{s} \log$ is positively monotonic, we can apply Jensen again to $\exp(E X) \le E \exp(X)$ and find
> > > > > > > $$ \frac{1}{s} \log \circ \exp (- s E_{k \sim \pi(\langle Z \rangle)} MMD(Z, k)) \le \frac{1}{s} \log E_{k \sim \pi(\langle Z \rangle)} [\exp \left( -s MMD(Z, k) \right)].$$
> > > > > > > Finally, we apply the bound from l.829, to find
> > > > > > > $$ -FUSE_1 \le \frac{1}{s} \log E_{k \sim \pi(\langle Z \rangle)} [\exp \left( -s MMD(Z, k) \right)] \le_{1-\delta} \frac{4\kappa^2 s}{n(n-1)} + \frac{\log\frac{1}{\delta}}{s} $$
> > > > > > > Which gives the overall result. Note that this final inequality is a $1-\delta$ high probability statement arising from the use of Markov’s inequality in l.829 first inequality.
> > > > > > >
> > > > > > > This is essentially a more detailed form of the proof while using $f(Z, k) = MMD(Z, k)$.
> > > > > > > The result for $FUSE_N$ follows by instead setting $f(Z, k)$ to the normalised MMD.

---

> > > > > > > > ### Comment · Reviewer_xB7K · 2023-08-18
> > > > > > > >
> > > > > > > > I see, thank you for clarifications. I would suggest to show the bound for optimal t in Theorem 5 in the final version, I think it looks more clear.
> > > > > > > >
> > > > > > > > Congratulations on your paper!

---

> > > > > > > > > ### Author Response · Authors · 2023-08-21
> > > > > > > > >
> > > > > > > > > Thank you for your time and attention to detail in our proofs, we really appreciate the feedback and suggestions that will make the proofs more easily understandable for the readers!

---

### Official Review · Reviewer_1tja · 2023-07-06

**Soundness:** 4 excellent
**Presentation:** 2 fair
**Contribution:** 4 excellent
**Rating:** 7
**Confidence:** 5

**Summary:**

This paper proposes a new statistic that can incorporate with the permutation test where we can learn features of data in an unsupervised way. From the perspective of testing, this paper contributes a new idea and paves a new way to perform two-sample testing without data splitting. If we can safely use all data to find a proper kernel, the test power will increase for sure. Theorems and proofs also verify the proposed methods.

**Strengths:**

1. This paper focuses on an important problem: two-sample testing, which will be very important when generators are like humans nowadays.

2. The selected research direction, testing without data splitting, is very important. As demonstrated in summary, a proper kernel selection in this direction can directly boost the test power.

3. Experiments include simple data and complex data (Galaxy MNIST and CIFAR), verifying the effectiveness of the proposed statistics.

**Weaknesses:**

1. The presentation can be improved a lot. The major issue is the lack of motivation for many choices. For example, what is the motivation to consider a mean kernel? what is the motivation to have this kind of design: log( E( exp( ) ) )? Can we consider more possibilities? The finding of this paper is exciting, however, it looks like a rush version. A revised version with a better presentation is required during the rebuttal.

2. Section 3 is a little bit long. The authors can consider introducing the theorem in the preliminary (as it is not proposed in this paper). Or the authors can remove the permutation test from Section 2. You can also consider merging Sections 2 and 3, making demonstrations more compact.

3. After removing the necessary parts in Sections 2 and 3, more experimental results can be put into the main body. For example, CIFAR results look very promising and should be moved into the main body.

4. In line 115, there is a notation typo.

5. Marks are recommended in Figure 1.

**Questions:**

Please address the points in Weakness. The presentation can be improved to make this paper even better.

---

> ### Author Rebuttal · Authors · 2023-08-08
>
> *[Comment to all reviewers (Author Rebuttal) posted above]*
>
> Thank you for your valuable feedback, for highlighting the strengths of our paper, as well as ways in which we can present and motivate our ideas more clearly. We will include the above discussions into the body of the paper.
>
> &nbsp;
>
> > what is the motivation to consider a mean kernel?
>
> The mean kernel simply arises naturally when considering distributions over kernels: since the squared MMD is linear, the expectation of it with respect to the kernel is equivalent to a single MMD^2 with a mean kernel (c.f. discussion l.238-243). When considering a fixed finite set of kernels of size $|K|$ with uniform prior as in our experiments, the equation on l.243 gives
>
> $$
> \widehat{\mathrm{FUSE}}_1(Z) = \sup^\rho\ \mathbb{E}^\rho[\widehat{\mathrm{MMD}}^2(X,Y; k)]  - \mathrm{KL}(\rho,\pi) / \lambda
> $$
> ($\rho$ should be a subscript but this syntax does not work)
>
> $$
> \widehat{\mathrm{FUSE}}_1(Z) = \sup_w\ \sum_i w_i\widehat{\mathrm{MMD}}^2(X,Y; k_i)  - \sum_i w_i \log(w_i |K|) / \lambda
> $$
>
> which shows that the $\widehat{\mathrm{FUSE}}_1$ statistic is just optimising the weightings of these kernels.
>
> We also illustrate in Appendix B that under certain choices (e.g. Gaussian kernel with a Gamma prior), the mean kernel can be expressed in closed form and corresponds to some commonly-used kernels (e.g. rational quadratic). This can theoretically enable continuous optimisation of the parameters of such mean kernels, while all previous methods required a discretisation of the parameter space.
>
>
> &nbsp;
>
> > what is the motivation to have this kind of design: log( E( exp( ) ) )? Can we consider more possibilities?
>
> Using the maximum directly is possible but would lead to a Bonferroni-type $\log(|K|)$ term in the power, and be limited to a finite number of “base” kernels. Also, only information from the single maximising “base” kernel can be considered at a time. For these reasons we use a “soft” maximum, which avoids the Bonferroni correction when more than one of the kernels is good, and considers information from every kernel simultaneously.
>
> Other soft maxima (see eq. 3) could certainly be considered, but the logsumexp variant we use is by far the most common choice. This is because it makes obtaining exponential concentration inequalities (by bounding the MGFs; see Theorem 10 in Appendix D) possible, and because the dual formulation (eq. 2 and l.243) allows us to derive power results in terms of MMD directly instead of in terms of the FUSE statistic.
>
> &nbsp;
>
> > The authors can consider introducing the theorem in the preliminary (as it is not proposed in this paper). Or the authors can remove the permutation test from Section 2. You can also consider merging Sections 2 and 3, making demonstrations more compact.
>
> We will work to remove unnecessary redundancy in these sections.
>
> > After removing the necessary parts in Sections 2 and 3, more experimental results can be put into the main body. For example, CIFAR results look very promising and should be moved into the main body.
>
> Thank you for highlighting the relevancy of this experiment - we will include it in the main body.
>
> > In line 115, there is a notation typo.
>
> We will add a definition for the notation $X =^d Y$ (i.e. $X$ and $Y$ follow the same distribution).
>
> > Marks are recommended in Figure 1.
>
> Thanks for pointing this out, we will add this.

---

> > ### Comment · Reviewer_1tja · 2023-08-16
> > **My concerns are addressed well.**
> >
> > Thanks for addressing my concerns. It is necessary to revise the paper to increase its readability for future readers.

---

> > > ### Author Response · Authors · 2023-08-21
> > >
> > > We are happy to hear your concerns have been addressed. Thank you for the feedback provided, we believe that incorporating it in the final version will greatly improve the readability of the paper.

---

### Author Rebuttal · Authors · 2023-08-08

# Comment to all reviewers

We warmly thank all reviewers for their time and detailed commentary on our work. We are highly encouraged by the positive evaluations of all reviewers: **reviewer 1tja** points out that our FUSE method *'paves a new way to perform two-sample testing without data splitting'*; **reviewer fFeS** that *'the delivered experiments are done very thoroughly and structured and the code is provided in a high quality*'; and **reviewer xB7K** that we prove *'concentration results for the test both under null and alternative'* which as noted by **reviewer jkkz** allows us to *'justify [our] method by showcasing the optimal MMD separation rate'*.

We appreciate the insights and feedback provided which can greatly improve the presentation of our ideas. Individual answers to reviewer questions are given below, and will be incorporated into the paper. We hope these address any remaining concerns raised by the reviewers, and if so, that they would kindly consider upgrading their evaluation scores.


# Additional Motivation for our proposed statistics and work
This is particularly relevant to reviewer fFeS.
We will add the following additional simple motivation for FUSE (different from the one presented in Section 4.1) to our work.

Say we are interested in two-sample testing, and know the MMD is a useful kernel measure in that setting, so using all the data available we decide to compute some MMD values
$$
\widehat{\mathrm{MMD}}_1,\ \dots\ ,\widehat{\mathrm{MMD}}_L
$$
for various kernels $k_1,\dots,k_L$ (finitely many for simplicity). A natural question is then 'What can we do with these values?' One possibility is to perform multiple testing as done in the case of MMDAgg. However, an even simpler and more intuitive answer would be to simply take the maximum of those values
$$
\max_i \widehat{\mathrm{MMD}}_i
$$
since after all the aim is to detect differences between the distributions (if they exist).

Now, there are two issues.

Firstly, this would lead to a Bonferroni-type $\log(|K|)$ term in the power (see discussion below), and be limited to a finite number of “base” kernels. Also, only information from the single maximising “base” kernel can be considered at a time. For these reasons we use a “soft” maximum, which avoids the Bonferroni correction when more than one of the kernels is good, and considers information from every kernel simultaneously. Other soft maxima (see eq. 3) could certainly be considered, but the logsumexp variant we use is by far the most common choice. (see also discussion in response to Reviewer 1tja).

The second issue is that, as you point out, the MMD values might have different scales and hence cannot be compared to each other meaningfully. Simply requiring that the kernels integrate to one is not enough, as the MMD estimates with different kernels can have varying variances which need to be accounted for. For this reason, in order to be able to meaningfully compare MMD values between each other, these need to be normalised before the maximum is taken. FUSE1 is a first step towards the idea of using soft maxima, but it does not solve the scaling problem, so FUSE_N is effectively a “studentised” variation, dividing through by a variance-like term, which is a common approach as in Sutherland et al. (and others, see l.157-158).

The specific form of the normalizer is motivated by the following. In order to make our statistic perform well we would like its null distribution to be sub-Gaussian-like (i.e. sub-Gaussian for optimal settings of lambda). From theorem 7, normalising by a term at least as big as $\bar{U}$ will achieve this, but this term is not permutation invariant and so would lead to large additional computation. We therefore replace it with a permutation-invariant upper bound. This gives an overall statistic FUSE_N which can be efficiently computed, but has tight null concentration leading to good power.


### Motivation versus MMDAgg

MMDAgg is a different way to think about combining multiple kernels and MMD values, but it relies on a framework based on multiple testing.
By contrast, our proposed FUSE method avoids multiple testing: a single statistic and quantile are used.

This can be beneficial in the case where the number of kernels considered increases (is large).
In that setting, the multiple testing approach MMDAgg combines more and more kernels, the level needs to be corrected differently depending on the number of kernels, and it is not theoretically guaranteed how the power of MMDAgg behaves in that setting (i.e. it could maybe go to zero).
We do not suffer from such issues with FUSE as even when increasing the number of kernels we still have only one statistic and quantile.
The FUSE statistic is even defined in the limit of an infinite number of kernels (continuous/uncountable collection of kernels) by considering distributions on the space of kernels, which is clearly not the case for MMDAgg (only defined for discrete collection of kernels).
Moreover, while the MMDAgg approach is only useful for hypothesis testing, having a quantity like FUSE combining multiple kernel-based measures of distance with exponential concentration bounds could be of interest in a wide range of applications.

---

### Decision · Program_Chairs · 2023-09-21

**Decision:**

Accept (spotlight)

**Comment:**

Reviewers agree that the paper significantly contributes to an important problem and includes convincing experiments.

During the rebuttal and discussion period, the authors undertake to improve the initial manuscript by including many clarifications, notably concerning the motivations of the proposed approach, the theoretical results, the proofs, and extended numerical results. It is essential to carefully include all of these in the camera-ready version of the paper.